# The role of carbon catalyst coatings in the electrochemical water splitting reaction

William J. V. Townsend[1,2], Diego López-Alcalá [3], Matthew A. Bird [1], Jack W. Jordan[1], Graham A. Rance [4], Johannes Biskupek [5], Ute Kaiser [5], José J. Baldoví [3], Darren A. Walsh [1,6] ✉, Lee R. Johnson [1,6] ✉, Andrei N. Khlobystov [2] ✉ & Graham N. Newton [1,6] ✉

Designing inexpensive, sustainable, and high-performance oxygen-evolution reaction (OER) electrocatalysts is one of the largest obstacles hindering the development of new electrolyzers. Carbon-coated metal/metal oxide (nano) particles have been used in such applications, but the role played by the carbon coatings is poorly understood. Here, we use a carbon-coated catalyst comprising metal-oxide nanoparticles encapsulated within single-walled carbon nanotubes (SWNTs), to study the effects of carbon coatings on catalytic performance. Electrolyte access to the encapsulated metal oxides is shut off by plugging the SWNT ends with size-matched fullerenes. Our results reveal that the catalytic activity of the composite rivals that of the metal oxide, despite the fact that the metal oxides cannot access the bulk electrolyte. Moreover, the rate-determining step (RDS) of the OER matches that measured at empty SWNTs, indicating that electrocatalysis occurs on the carbon surface. Synergism between the encapsulated metal oxide and carbon coating was explored using electrochemical Raman spectroscopy and computational analysis, revealing that charge transfer from the carbon host to the metal oxide is key to the high electrocatalytic activity of carbon in this system; decreasing electron density on the carbon surface facilitates binding of $^-$OH, accelerating the rate of the OER on the carbon surface.

Splitting water electrochemically to produce so-called "green hydrogen" requires identification of sustainable, robust electrocatalysts that can replace the unsustainable, expensive noble-metal catalysts used in state-of-the-art devices[1]. Electrolysis of water in electrolyzers proceeds through the oxygen evolution reaction (OER) at the positive electrode and the hydrogen evolution reaction (HER) at the negative electrode[2,3]. The OER is a relatively slow process that requires several complex electron- and proton-transfer reactions, lowering the efficiency of the overall water-splitting process[4]. Moreover, the stability and lifetime of

typical electrocatalysts are often relatively poor, due to the harsh oxidizing conditions at the positive electrode[5–9]. State-of-the-art electrocatalysts for the OER comprise rare and expensive metals (or their oxides), such as Pt, $RuO_2$, and $IrO_2$, and much effort has been devoted to designing electrocatalysts that can minimise the quantity of expensive metals, maximise their stability, or avoid their use completely[10–12]. The use of nano-structured electrocatalysts supported on conducting carbons can lead to enhanced stability and activity, and has allowed the use of earth-abundant first row transition metals, such

[1]Nottingham Applied Materials and Interfaces (NAMI) Group, GSK Carbon Neutral Laboratories for Sustainable Chemistry, School of Chemistry, University of Nottingham, Nottingham, UK. [2]School of Chemistry, University of Nottingham, Nottingham, UK. [3]Instituto de Ciencia Molecular (ICMol), Universidad de Valencia, Valencia, Spain. [4]Nanoscale and Microscale Research Centre, University of Nottingham, Nottingham, UK. [5]Electron Microscopy Group of Materials Science, Ulm University, Ulm, Germany. [6]The Faraday Institution, Didcot, UK. ✉e-mail: darren.walsh@nottingham.ac.uk; lee.johnson@nottingham.ac.uk; andrei.khlobystov@nottingham.ac.uk; graham.newton@nottingham.ac.uk

as Co, Fe and Ni, with activities comparable to those of traditional electrocatalytic materials[4,13–22].

A particularly powerful emerging approach to improving the activity and stability of electrocatalysts is to encapsulate them within carbon shells[23–27], and a similar strategy has been used for the development of single-layer hexagonal boron nitride (hBN)-coated electrocatalysts[28,29]. However, this approach can introduce uncertainty in terms of whether the active site is the encapsulated metal or the coating itself. Most researchers favour increasing the porosity of the carbon support to increase accessibility of the encapsulated metal centres, assuming that they are the active sites[30–36]. Others opt for modulation of the electronic structure of the support (as first proposed by Deng et al.[37]), under the assumption that the electrocatalysis is happening at the carbon shell surrounding the metal centre[22,38–41]. It has been suggested that charge transfer between the encapsulated species and the carbon coating tunes the work function of the carbons near the encapsulated material[37,42], as well as that of neighbouring carbon atoms[37], an effect similar to that observed by elemental doping of the carbon lattice[43]. Electronic modulation of the carbon can be directed towards electrocatalytic oxidations or reductions by controlling the magnitude and direction of the charge transfer[42]. Computational work has also indicated that a single atomic layer of carbon gives the largest charge transfer effect and that the improvement drops off dramatically as more layers are added[37,44]. Charge transfer can also influence the electronic structure and electrocatalytic activity of the metal surfaces. The high activity of a Pt/C electrocatalyst has been attributed to electron transfer from a Pt d orbital to the carbon π* orbital, decreasing the water dissociation energy at the electron-deficient Pt[45–47]. However, examples in which the metal and carbon both provide active sites have also been proposed[48]. Computational studies also disagree on which is the active site; Li et al. proposed that iron oxide confined in SWNTs, rather than the carbon surface, was the active site for OER[49], while Ma's calculations suggested that $O_2$ reduction occurred on the carbon support of a carbon-Fe composite[50]. Prato and co-workers recently pointed out that more experimental efforts should be made to identify the active sites in such processes and that one should not assume that one site is more active than another[51].

In this work, we have prepared a series of model electrocatalytic OER systems based on nanoparticulate metal oxide ($MO_x$) electrocatalysts ($Co_3O_4$, $RuO_2$ and $IrO_2$) immobilized within pristine SWNTs, to identify the active site during electrocatalysis of the OER at carbon-coated electrocatalysts. Redox-active materials have been encapsulated within SWNTS[52] (as well as multi-walled nanotubes[37,53–55]) previously, and used for electrocatalytic reactions such as the oxygen reduction reaction (ORR) and HER[56,57]. However, the active sites of electrocatalysis in these systems were not experimentally verified. We have designed a system in which the electrocatalysts are completely separated from the external electrolyte, not only by the sidewalls of the SWNTs but also by plugging the open ends of the SWNTs with size-matched fullerenes. Comparing the activity of the nanocomposite with that of empty SWNTs unambiguously reveals that the active electrocatalytic site is the carbon coating the electrocatalysts. However, the electrocatalytic behaviour is similar to that expected at a bare $MO_x$ surface, despite the fact that the reaction occurs on the surface of a single layer of carbon atoms. Spectroscopic and electrochemical analyses, paired with computational studies, show that charge transfer from the carbon host to the encapsulated metal oxide is key to the operation of these materials. It is important to note that we do not expect that these "$MO_x$@SWNT" materials are likely to become practical electrocatalysts for devices, rather our work demonstrates that modulation of the electronic structure of carbon can drastically affect its electrocatalytic behaviour. Such electronic modification of carbons could potentially be a new approach towards the development of a new generation of sustainable carbon-based

catalysts with activities similar to those of transition metal based electrocatalysts.

## Results and Discussion

Arc-discharge SWNTs with a narrow diameter distribution of 1.4-1.5 nm and lengths of over 1 micron were chosen as hosts for the electrocatalysts. The SWNTs have low defect densities and small diameter distribution, which is essential when using bulk electrocatalytic activity to rationalise a mechanistic pathway and to limit access of the electrolyte into the inner pores and thus to the surface of the metal oxides. An important step in preparing our nanocomposite materials was the removal of the SWNT end caps and residual Ni catalyst that could contribute to catalytic activity[58,59]. This was achieved by refluxing the SWNTs in 3.0 M $HNO_3$, followed by heating in air and a wash in concentrated HCl (Fig. 1a), yielding metal-free, open-ended SWNTs. Thermogravimetric analysis (TGA) gave a residual weight of <0.2 % after combustion, showing that SWNTs were essentially free from Ni (Supplementary Fig. 1). This was corroborated by the absence of Ni signals during X-ray photoelectron spectroscopy (XPS) (Supplementary Fig. 2). Encapsulation of metal oxides in SWNTs was achieved by subliming the corresponding metal carbonyls ($M_x(CO)_y$) into the SWNTs under reduced pressure, followed by oxidation in air. The relative ease of sublimation and small molecular diameter of metal carbonyls make them attractive precursors for filling 1.4-1.5 nm diameter SWNTs (Fig. 1b–d)[60–62]. To obtain $IrO_2$ and $RuO_2$, the relevant carbonyl/SWNT materials were heated in air to 170 °C and 150 °C, respectively, decomposition temperatures that were based on relevant TGA (Supplementary Fig. 3). Exposure of the $Co_2(CO)_8$/SWNT hybrid to air at room temperature led to spontaneous oxidation to $Co_3O_4$, permitting isolation of a $Co_3O_4$/SWNT material, without the need for any additional reaction steps[63]. The syntheses yielded composites in which metal oxides were located inside the SWNTs and adsorbed on the exterior surfaces. The material adsorbed on the SWNT exteriors was removed by washing (Experimental section and Supplementary Fig. 4), yielding $MO_x$@SWNT materials, in which the metal oxides reside within the SWNT cavities. TGA-determined mass loadings of metal oxides were 6, 13 and 16% for $Co_3O_4$@SWNT, $RuO_2$@SWNT and $IrO_2$@SWNT, respectively (Supplementary Figs. 1 and 3), corresponding to similar atomic percentages of each metal in the composite materials (Supplementary Table 1).

High-resolution transmission electron microscopy (HRTEM) imaging was performed on the $MO_x$@SWNT materials to characterize the encapsulated metal oxides (Fig. 1b–d, Supplementary Figs. 4–8, Supplementary Note 1). Confined metal-oxide particles appeared throughout each sample, in the form of nanorods with lengths of 1-10 nm and a width of about 1 nm. Due to the confinement in the SWNT interior, the width of $MO_x$ was restricted, ensuring that the encapsulants were in direct contact with the SWNT sidewalls, and the internal cavity of the nanotube was effectively blocked. This is in marked contrast to the structure of previously-reported OER electrocatalysts immobilized inside wider MWNTs, to which the external electrolyte had access through the open nanotube channels[54]. Energy-dispersive X-ray spectroscopy (EDX) mapping was performed on these materials during TEM imaging, which revealed characteristic Ir, Ru and Co peaks for $IrO_2$@SWNT, $RuO_2$@SWNT and $Co_3O_4$@SWNT, respectively (Supplementary Fig. 9). EDX mapping showed a uniform distribution of carbon, oxygen and the corresponding metal for each of the $MO_x$@SWNT materials, confirming that the encapsulated metals were present throughout the sample. These data are further supported by XPS analysis confirming the expected elemental composition of each catalyst (Supplementary Fig. 10).

XPS of the $MO_x$@SWNT materials was performed to elucidate the properties of the carbon and the chemical states of the confined $MO_x$ (Fig. 1e and Supplementary Figs. 10 and 11). To accurately compare the position of the carbon 1s peak, the samples were loaded onto silicon

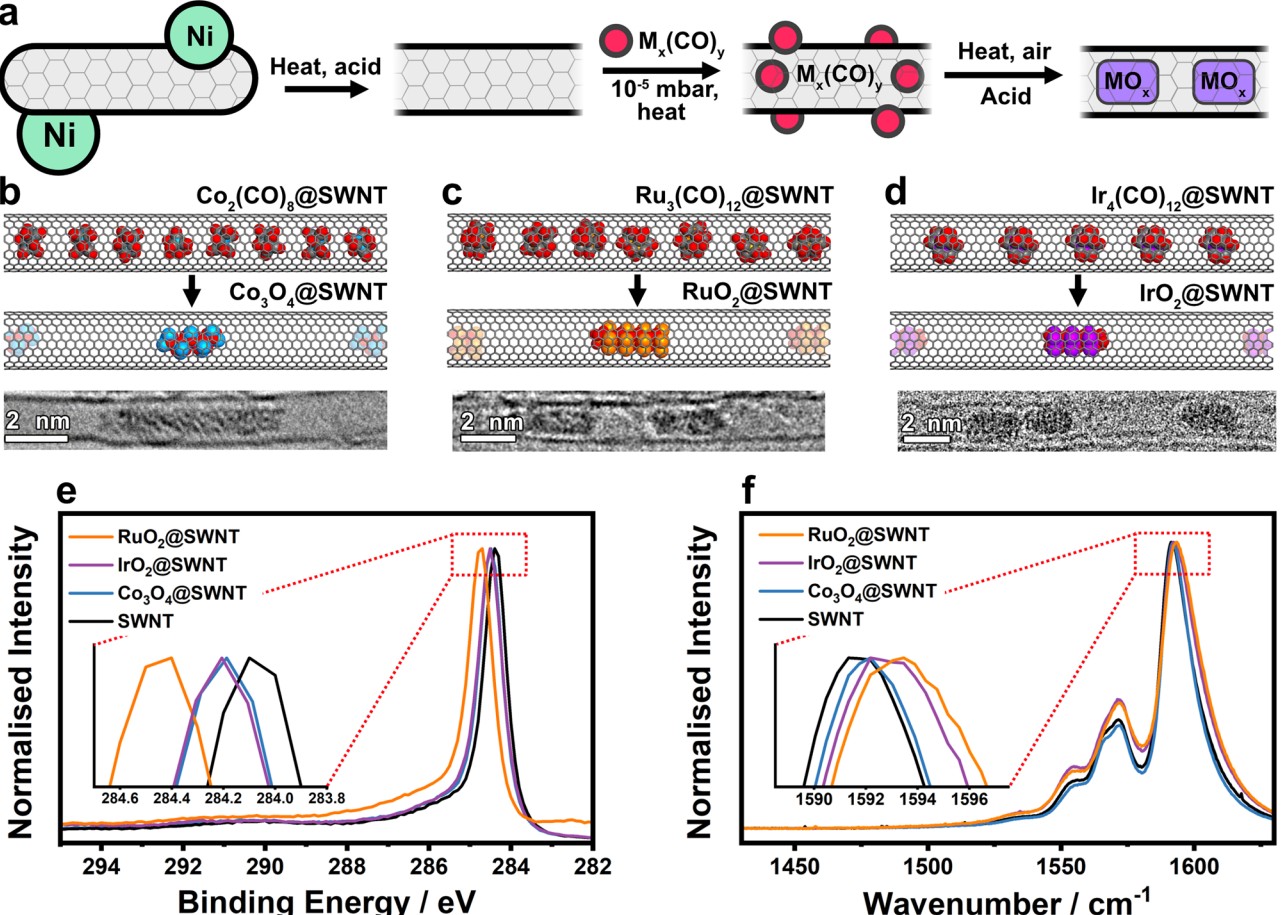

**Fig. 1 | Encapsulation of metal oxides in SWNTs. a** Scheme showing the general synthetic route for MO$_x$@SWNT. The Ni catalyst is removed by washing and the metal carbonyl precursors are inserted into the SWNTs and decomposed in air to the corresponding MO$_x$. Schemes showing the decomposition of **b** Co$_2$(CO)$_8$@SWNT to Co$_3$O$_4$@SWNT, **c** Ru$_3$(CO)$_{12}$@SWNT to RuO$_2$@SWNT and **d** Ir$_4$(CO)$_{12}$@SWNT to IrO$_2$@SWNT. The relative amounts of precursors and MO$_x$ compared to carbon in **b–d**, are estimated from the TGA data (Supplementary Fig. 1) and are shown in a 10-nm segment of a SWNT. An AC-TEM image of Co$_3$O$_4$@SWNT and HR-TEM of RuO$_2$@SWNT and IrO$_2$@SWNT is shown below each schematic. **e** X-ray photoelectron spectra showing the C 1 s region of RuO$_2$@SWNT, IrO$_2$@SWNT, Co$_3$O$_4$@SNWT and SWNT. Inset shows the highlighted peak maxima. Wide-scan spectra of each material can be found in the SI (Supplementary Fig. 10). **f** Raman spectra showing the G band of RuO$_2$@SWNT, IrO$_2$@SWNT, Co$_3$O$_4$@SNWT and SWNT. Inset shows the highlighted peak maxima. Wide-scan spectra of each material can be found in the SI (Supplementary Fig. 12).

wafers, and the Si 2$p$ peak at 99.5 eV[64] was used as an internal reference to correct for charge accumulation (Supplementary Fig. 10). The carbon peaks of all MO$_x$@SWNT were shifted to a higher binding energy compared to those of unfilled SWNTs, with RuO$_2$ displaying the largest shift. This shift suggests that electron density on the carbon surface decreased due to charge transfer to the encapsulated metal oxide, which is consistent with reports of similar materials[63]. Characteristic peaks of Co, Ru and Ir matching well to Co$_3$O$_4$, RuO$_2$ and IrO$_2$, respectively, were also observed (Supplementary Fig. 11, Supplementary Note 2).

Raman spectra of SWNTs show characteristic peaks in the regions of ~1350 cm$^{-1}$ (D band) and ~1590 cm$^{-1}$ (G band) that correspond to vibrations within disordered and ordered ($sp^2$) carbon domains (Supplementary Fig. 12). The ratio of the intensities of these peaks (I$_D$:I$_G$) can be used as a means to quantify the density of defects in SWNTs[65]. All samples measured had low levels of structural defects, with calculated densities <12 defects μm$^{-1}$ (Supplementary Table 1), demonstrating that the experimental conditions during encapsulation of metal oxides led to only small increases in the number of defect sites, and accordingly that the encapsulated metal oxide sites were effectively separated from the electrolyte by the intact SWNT walls. The G-band positions in the spectra of all

MO$_x$@SWNT were blueshifted relative to that of unfilled SWNT (Fig. 1f), with RuO$_2$@SWNT displaying the largest shift, followed by IrO$_2$@SWNT and then Co$_3$O$_4$@SWNT. These shifts were in line with XPS analysis and further evidence of charge transfer from the SWNT to the encapsulated metal oxides[66].

The electrocatalytic activity of the MO$_x$@SWNT materials was compared to that of SWNTs using rotating-disk electrode (RDE) voltammetry in O$_2$-saturated aqueous 0.1 M KOH (Fig. 2a), conditions that are in line with those in previous studies (see Supplementary Information for details)[67–69]. While not designed to be 'practical' electrocatalysts, these systems allow us to isolate the specific role played by the carbon host when in close contact with metal oxide electrocatalysts. We selected alkaline rather than acidic conditions as a starting point for this study based on our recent observation that protons can transport readily through the graphenic walls of SWNTs[70], potentially complicating the electrochemical reaction mechanism. As a first assessment of the OER electrocatalytic activity, the overpotentials at which the current density reached 10 mA cm$^{-2}_{geo}$ ($\eta_{10}$) in 0.1 M KOH were compared (Fig. 2a)[11]. The activities of all MO$_x$@SWNT materials were greater than that of pristine SWNTs and in line with or exceeding previous reports of Co$_3$O$_4$[71,72], RuO$_2$[7,73,74], and IrO$_2$[7,75], despite the fact that the electrocatalysts were encapsulated within the SWNTs.

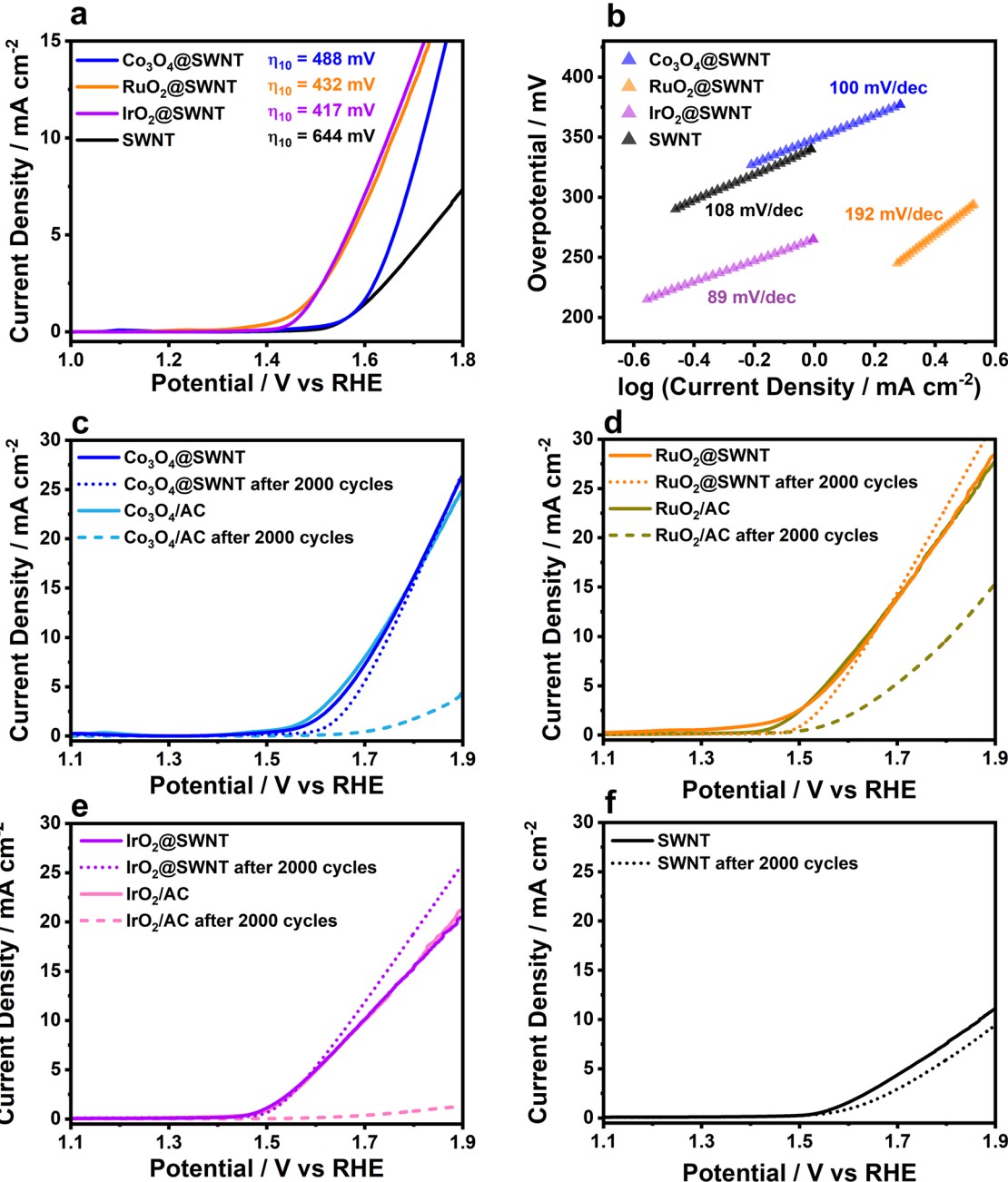

**Fig. 2 | Electrochemical performance of the electrocatalyst towards the OER. a** Rotating disk linear-sweep voltammograms (RDE-LSVs) of 0.5 mg cm$^{-2}$ Co$_3$O$_4$@SWNT, RuO$_2$@SWNT, IrO$_2$@SWNT and empty SWNTs recorded between 1.0 V and 1.8 V vs RHE at 1600 rpm and 5 mV s$^{-1}$ in O$_2$-saturated 0.1 M KOH. The test materials were immobilised on a glassy carbon (GC) disc electrode, and a graphite rod was used as the counter electrode. Currents are normalised to the geometric area of the disc. A Hg/HgO reference electrode was used in the cell, and all potentials were converted to the reversible hydrogen electrode (RHE) scale. **b** Plot of log current density vs overpotential for the OER, extracted from the LSVs shown in **a**. Tafel slopes were fitted from the most linear parts of the curves, as determined by R$^2$ value, with all Tafel slope values presented having R$^2$ values > 0.9995. **c**–**f** RDE-LSVs of MO$_x$@SWNT, MO$_x$/AC and empty SWNTs in N$_2$-saturated 0.1 M KOH. The materials were cycled between 0.9 and 1.7 V vs RHE at a scan rate of 100 mV s$^{-1}$, and voltammograms were recorded before and after, using a scan rate of 5 mV s$^{-1}$.

Tafel analysis of the RDE voltammetry data (Fig. 2b and Supplementary Table 2) yielded OER Tafel slopes of about 100 mV dec$^{-1}$ for Co$_3$O$_4$@SWNT, IrO$_2$@SWNT and empty SWNTs, indicating that the rate-determining step (RDS) was the same in all three systems. The values obtained are close to those obtained when binding of $^-$OH ions to the active site is the RDS[76], as reported for SWNTs in alkaline conditions[77]. The OER Tafel slope recorded using RuO$_2$@SWNT was higher than the others, due to the occurrence of a RuO$_2$ oxidation process at the onset of OER (Supplementary Fig. 13). Notably, these Tafel slopes are higher than those typically recorded during the OER at metal oxides, which are ~40-55 mV dec$^{-1}$[178–80], suggesting that the active sites in the MO$_x$@SWNT materials and SWNTs were the same (that is, the carbon surface). This observation is especially notable considering that the MO$_x$@SWNT materials still demonstrated metal oxide-like electrocatalytic activity. While our model systems are not expected to be suitable for application in commercial electrolysers, we explored the electrochemical stability of the carbon surface during potential cycling; carbon-coatings are frequently proposed to improve electrocatalyst stability and limit leaching of active materials[81]. Co$_3$O$_4$@SWNT, RuO$_2$@SWNT and IrO$_2$@SNWT were cycled 2000

times between 0.8 and 1.7 V (*vs.* RHE) in $N_2$-saturated 0.10 M KOH at 100 mV s$^{-1}$. RDE voltammetry of $Co_3O_4$@SWNT and $RuO_2$@SWNT before and after cycling (Fig. 2c–f) showed almost no change in the electrocatalytic behaviour, while $IrO_2$@SWNT exhibited a decrease in $\eta_{10}$ after cycling (Supplementary Table 2). The OER Tafel slopes decreased to 83, 67 and 77 mV dec$^{-1}$ for $Co_3O_4$@SWNT, $RuO_2$@SWNT and $IrO_2$@SNWT, respectively, after 2000 cycles (Supplementary Table 2). These changes are likely induced by partial loss of carbon from the SWNT walls leading to exposure of $MO_x$ to the electrolyte or by the introduction of oxygen-containing groups on the surface of the SWNTs during OER, which could lead to a change in local electron density on the carbon surface. This local charge redistribution on the carbon surface can lead to a higher binding energy for $^-OH$[43]. The stability of the empty SWNTs was also tested (Fig. 2f), and a large decrease in activity after cycling was observed, which was also likely due to carbon oxidation and loss. However, Raman spectroscopy of the materials before and after cycling (Supplementary Fig. 14) revealed less than 20 defects µm$^{-1}$ in the SWNTs (Supplementary Table 1), while inductively Coupled Plasma (ICP) analysis of all electrolytes showed no detectable leaching of metal ions after 2000 cycles.

For comparison with our $MO_x$@SWNT materials, $Co_3O_4$, $RuO_2$ and $IrO_2$ were also deposited onto activated carbons to yield $MO_x$/AC (see Supplementary Information for details), and OER activities and stabilities were measured (Fig. 2c–e, Supplementary Fig. 15). The initial activities of the $MO_x$/AC controls were comparable to those of the $MO_x$@SWNT analogues, but their activities decreased significantly over 2000 cycles. In addition, commercial $MO_x$ catalysts and $MO_x$ deposited onto closed SWNTs ($MO_x$/C-SWNT) were studied as controls (Supplementary Figs. 15–17) and OER activities were similar or lower than those of the $MO_x$@SWNT materials. Therefore, encapsulation of metal oxides within SWNTs appears to protect the confined species in the $MO_x$@SWNT materials. Moreover, the $MO_x$@SWNT systems exhibit high mass activities, which is consistent with the ultra-small size of the encapsulated metal oxides (Supplementary Fig. 16). To further test whether the carbon surface, and not the encapsulated metal oxide surface, was the active site, a series of OER measurements were run after the addition of KSCN to the electrolyte; KSCN binds to metal-oxide surfaces, limiting their electrocatalytic activity[82]. No change in the activity of the $MO_x$@SWNT materials was apparent, while KSCN caused a drastic decrease in the activity of the corresponding $MO_x$/AC materials, demonstrating further that the carbon surface was most likely the active site in the $MO_x$@SWNT materials (Supplementary Fig. 18).

Further confirmation that electrocatalysis was occurring at the carbon surface, rather than at exposed $MO_x$ was obtained by blocking any access of electrolyte to the encapsulated materials through the open ends of the SWNTs. This was done by "plugging" the open ends of the $MO_x$@SWNT with $C_{60}$ fullerenes (Fig. 3a, Experimental section)[70]. TEM images (Fig. 3b, Supplementary Fig. 19) of plugged $MO_x$@SWNT ($C_{60}$-$MO_x$@SWNT) show co-encapsulation of fullerenes within the free spaces at the ends of the SWNTs. The van der Waals diameter of $C_{60}$ matches the opening of these SWNTs, leaving no physical gap when $C_{60}$ is inserted, severely limiting the diffusion of electrolyte into their internal cavity (Fig. 3c–e)[83]. The voltammograms of $Co_3O_4$@SWNT recorded before plugging revealed a small oxidation peak at 1.11 V (Fig. 3f inset blue line), corresponding to oxidation of $Co_3O_4$[84]. The charge passed corresponded to oxidation of just 2.7% of the total encapsulated $Co_3O_4$ (assuming that the reaction is $Co_3O_4 + H_2O + ^-OH \rightarrow 3CoOOH + e^-$) and was likely that at the ends of the SWNTs. Once plugged, the charge passed decreased by 90% (Fig. 3f inset dotted blue line), confirming that direct access of the electrolyte to the oxide was almost completely blocked. A similar oxidation process was seen during voltammetry of $RuO_2$@SWNT and it was also completely blocked upon plugging the SWNTs (Supplementary Fig. 20). In the case of $IrO_2$@SWNT, this effect could not be discerned, as the

oxidation of $IrO_2$ occurred at more positive potentials, where the current from OER masked the process. It is important to note that some oxidation of the metal oxide was observed using ex-situ XPS after polarisation to 1.6 V (Supplementary Fig. 21), indicating oxidation of the blocked material could occur at higher overpotentials.

The OER activities of $MO_x$@SWNT before and after plugging the ends of the SWNTs were very similar, with an increase in $\eta_{10}$ of between 8 and 31 mV after plugging (Fig. 3f–h). This small increase in $\eta_{10}$ is likely due to a decrease in electrochemical surface area by $C_{60}$ adsorption into the SWNT network, as evidenced by a decrease in the capacitance by about ~5% after plugging (Supplementary Fig. 22). Our observation that the OER activity is unaffected by completely coating it with carbon unambiguously demonstrates that the active sites are on the carbon surface. Moreover, it is remarkable that the activity of all $C_{60}$-$MO_x$@SWNT exceeds that of pristine SWNTs, showing that the electrocatalytic sites on the carbon surface are directly enhanced by the encapsulated $MO_x$. Finally, to test whether fullerene encapsulation has an inherent effect on the OER activity of the carbon shell, we plugged pristine SWNTs with fullerenes. No significant change in OER activity was observed after plugging (Supplementary Fig. 23).

In-situ electrochemical Raman spectroscopy was performed to elucidate further the role of the encapsulated metal oxides on the increase in OER activity (Supplementary Fig. 24). The data for $Co_3O_4$@SWNT and SWNT is shown (Fig. 4) for clarity, while the data for $RuO_2$@SWNT and $IrO_2$@SWNT can be found in the Supplementary Information (Supplementary Figs. 25, 26 and Supplementary Table 3). The G bands in the spectra of SWNTs and $Co_3O_4$@SWNT blue shifted as a positive potential was applied (Fig. 4a), due to p-doping of the SWNTs[85]. However, the shift in the spectrum of $Co_3O_4$@SWNT was more significant, increasing by 12 cm$^{-1}$ over 1.0 V, compared to 7 cm$^{-1}$ for SWNTs. A larger blueshift over this potential range was also observed for $RuO_2$@SWNT and $IrO_2$@SWNT (Supplementary Fig. 26) compared to SWNTs. These positive shifts confirm that transfer of electron density from the carbon to the encapsulated $MO_x$ occurred, resulting in a more positive charge on the carbon surface. This transfer of charge from the carbon made binding of $^-OH$ more favourable, consistent with our Tafel analysis and the higher OER activity for all $MO_x$@SWNT materials. An increase in $I_D$:$I_G$ for the SWNT material occurred at potentials positive of 1.5 V (Fig. 4b), coinciding with the onset of OER (Supplementary Fig. 27) and consistent with the adsorption of $^-OH$ groups or formation of other oxygen-containing groups, disrupting the $sp^2$ character of the carbon[86]. This increase in $I_D$:$I_G$ occurred to a greater extent for $Co_3O_4$@SWNT (Fig. 4b, Supplementary Table 3), starting from an earlier onset potential of 1.3 V, consistent with the lower overpotential for OER for this material, and further demonstrating that the OER occurs at the surface of SWNTs. This increase in $I_D$:$I_G$ at a lower onset potential was also observed for $IrO_2$@SWNT and $RuO_2$@SWNT (Supplementary Fig. 26).

Charge transfer between the carbon and the encapsulated metal oxides was explored using first-principles DFT calculations as implemented in the CP2K/QUICKSTEP package[87,88] (see computational details in Supplementary Information). We modelled encapsulation of $Co_3O_4$, $RuO_2$ and $IrO_2$ fragments in (11,11)-SWNTs to reveal the changes in electron density when introducing a $MO_x$ section into the SWNT. Figure 4c shows the charge density difference after the encapsulation of $RuO_2$ (data for $IrO_2$@SWNT and $Co_3O_4$@SWNT are shown in Supplementary Figs. 28 and 29). A systematic charge transfer from the C π-electron system to the $MO_x$ can be observed in all cases and the transfer was predominantly to the outer oxygen atoms of the encapsulated metal oxides. As the terminal O atoms in the transition metal oxide are highly electronegative, they attract electron density from the delocalized π system on the inner part of the SWNT. The charge transfer was not necessarily homogenous, with the largest occurring on $MO_x$ oxygen atoms closest to the inner part of the SWNT (Supplementary Figs. 28 and 29). This demonstrates that the electronic

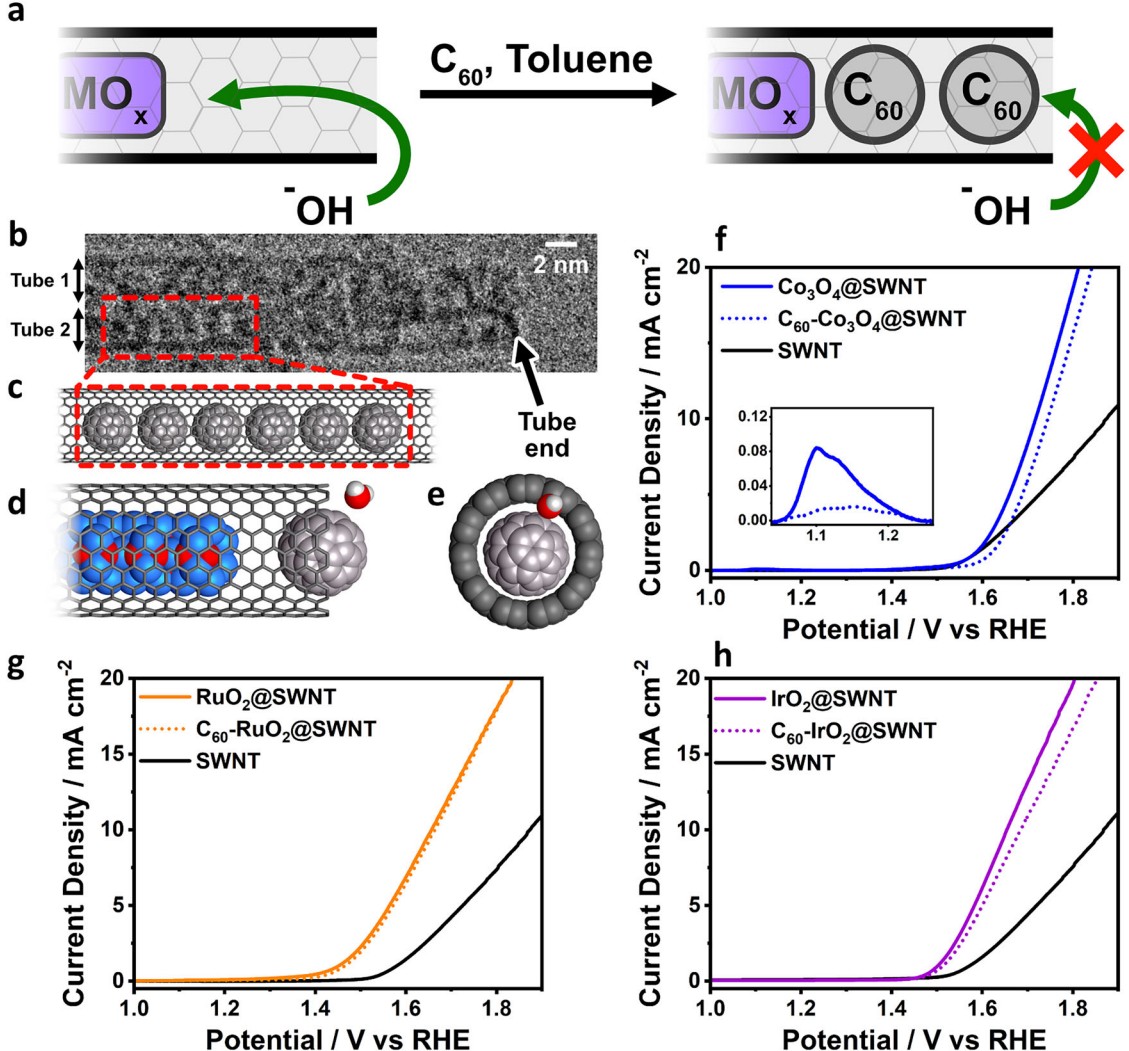

**Fig. 3 | Plugging Co₃O₄@SWNT with fullerenes. a** Scheme showing plugging of Co₃O₄@SWNT with C₆₀. **b** HR-TEM image showing fullerene filling at SWNT ends in the C₆₀-plugged Co₃O₄@SWNT composite. **c** Schematic illustration showing fullerenes filled into a SWNT. **d** and **e** Schematic illustration of C₆₀-Co₃O₄@SWNT, showing how fullerene encapsulation limits transport of $H_2O$ to the encapsulated Co₃O₄. **f–h** Rotating disc linear-sweep voltammograms (RDE-LSV) (at 1600 rpm, 5 mV s⁻¹) in N₂-saturated 0.1 M KOH of Co₃O₄@SWNT, RuO₂@SWNT and IrO₂@SWNT respectively, before and after C₆₀ plugging, as well as the activity of SWNT for comparison. The inset shown in (f) shows a Co₃O₄ oxidation around 1.1 V vs RHE, which is no longer accessible upon plugging.

structure and morphology of the encapsulated material affected charge transfer. In addition, differences in the charge-transfer resistance at OER potentials were observed (Supplementary Fig. 30) and depended on the identity of encapsulated $MO_x$.

Charge transfer in all systems was demonstrated quantitatively using Mulliken charge analysis (See computational details in the SI). RuO₂@SWNT showed the largest charge transfer value of 3.49 electrons (per RuO₂ section modelled as shown in Fig. 4c), followed by IrO₂@SWNT (2.83 electrons) and Co₃O₄@SWNT (1.04 electrons). The higher degree of charge transfer in RuO₂@SWNT and IrO₂@SWNT was mirrored in the Raman spectroscopy and XPS data (Fig. 1e and f) and the electrochemical behaviours of the materials (Fig. 2a); RuO₂@SWNT and IrO₂@SWNT showed larger shifts in G-band position and lower onset potentials for the OER. The calculated density of states of these $MO_x$@SWNT materials are also shown in Supplementary Fig. 31. Introduction of oxygen-containing groups on the carbon surface during a typical OER cycle (e.g -O and -OH) was also modelled and resulted in an even larger atomic charge on neighbouring carbon atoms (see Supplementary Fig. 32, Supplementary Table 4 and Supplementary Note 3), consistent with an increase in electrochemical activity and decrease in Tafel slope after electrochemical cycling (Fig. 2c–e, Supplementary

Table 2). In summary, the in-situ spectroelectrochemical studies and computational analysis support the experimental evidence that the active sites were on the carbon surface and that the increase in OER catalytic performance originates from charge transfer between the metal oxide and carbon. This charge transfer results in a depletion of electron density on the carbon surface, facilitating the binding of ⁻OH, which appears to be the RDS. Based on these analyses, our proposed catalytic cycle is shown in Fig. 4d. This is consistent with the adsorbate evolution mechanism (AEM) OER pathway in alkaline conditions[89] and previous reports on OER at SWNTs[90].

We have demonstrated a model carbon-coated electrocatalyst, in which metal oxide nanoparticles are fully encapsulated in a carbon shell. This configuration prohibited redox processes from occurring at the surface of the metal oxide and allowed us to study reactions at the carbon surface in isolation. In doing so, we have shown that by carbon coating the metal oxides, the inherent catalytic activity of the metal oxide was transferred to the carbon shell, where electrocatalysis occurred. The electronically-modified carbon surface showed metal oxide-like activity for the oxygen evolution reaction, yet electrochemical kinetic analysis indicates that the reaction mechanism remains that of OER at a carbon surface and not at the underlying

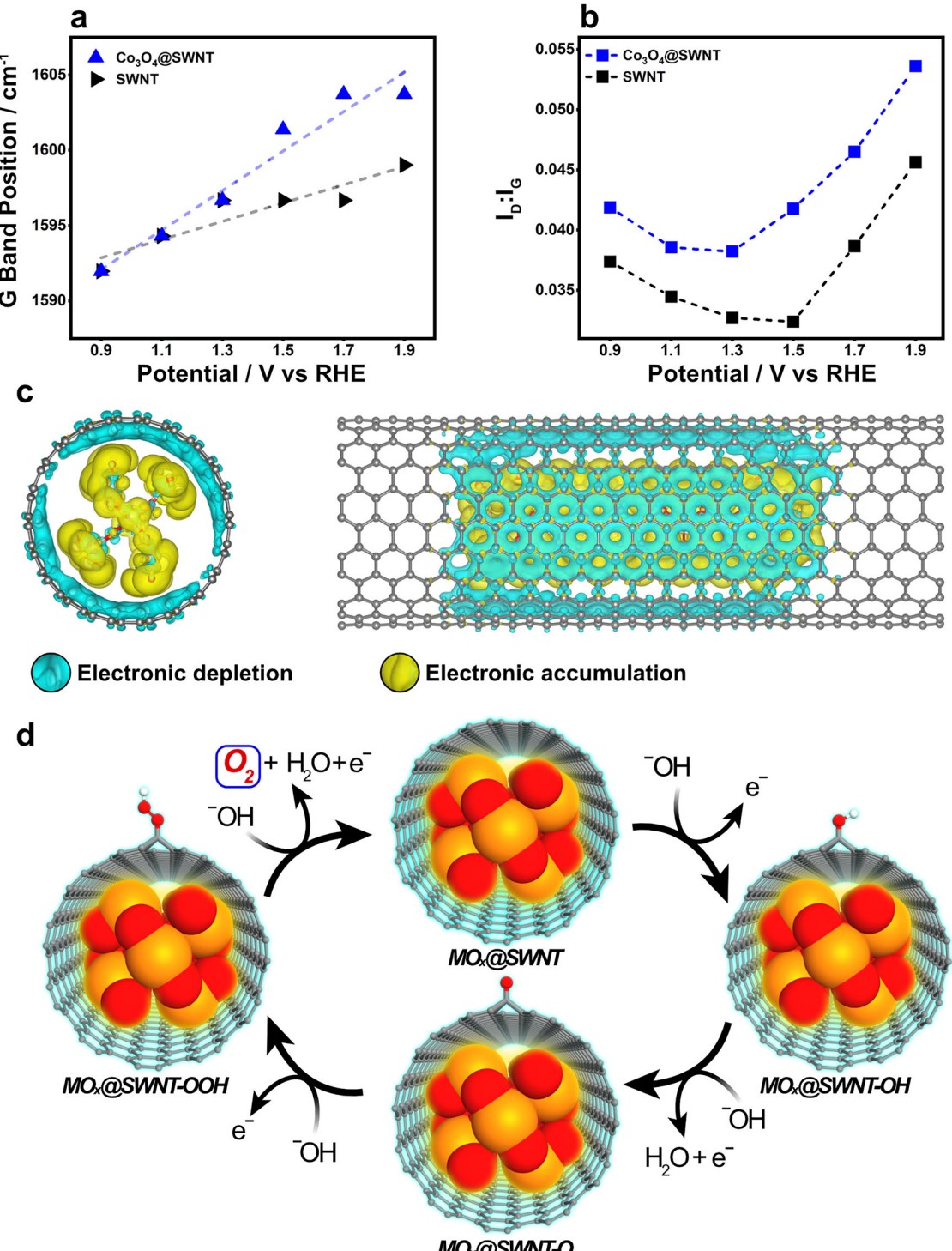

**Fig. 4 | Mechanistic Understanding of MO$_x$@SWNT for OER. a, b** G band positions and I$_D$:I$_G$ ratios respectively taken from the 785 nm Raman spectra of Co$_3$O$_4$@SWNT and SWNT. Spectra are taken at potential holds, from 0.9 V to 1.9 V to 0.9 V vs RHE from bottom to top in 0.2 V steps, in N$_2$ saturated 0.1 M KOH. Each data point shown represents the average of 9 spectra collected from a 50 × 50 μm regular array across the modified GC surface. **c** Electronic density difference after the encapsulation of RuO$_2$@SWNT (11, 11). Blue (yellow) colour represents the depletion (augmentation) of electronic density. **d** Proposed mechanistic cycle of oxygen evolution reaction on the carbon surface next to encapsulated metal oxide.

metal oxide. By exploring the electronic properties of the carbon surface spectroscopically and computationally, we attribute this enhanced catalytic activity to depletion of the electron density of the carbon shell by the underlying metal oxide. It is known that the rate-determining step during oxygen evolution at carbon is binding and oxidation of ⁻OH, and depleting the electron density of the carbon shell increases the oxophilicity of the carbon, promoting this step.

Coating electrocatalysts with carbons is an increasingly popular choice for improving electrocatalytic activity and one that is often misunderstood. These insights clarify the electrocatalytic mechanisms at these materials and open new design space for the development of high-performance, sustainable electrocatalysts by electronic modification of carbon coating, potentially through interaction with inexpensive redox-active materials. While we have shown the impact of

such effects using an oxidation reaction as the probe reaction, the materials approach and modulation of the carbon electronic structure is expected to be applicable to catalysis of a range of electrochemical and chemical reactions.

# Methods

### Materials

SWNTs (P2-SWNTs, arc discharge, Carbon Solutions, USA) were refluxed in nitric acid (3.0 M) for three hours and filtered with deionised water, they were then heated at 600 °C for 30 minutes to open the SWNT termini, giving a 50 % weight loss. Finally, they were sonicated for 30 minutes in concentrated HCl, filtered and washed with deionised water until a neutral pH was shown. All other chemicals were used as supplied (Sigma Aldrich).

### $Co_3O_4$@SWNT

$Co_2(CO)_8$ (20 mg, 0.058 mmol) and freshly opened SWNTs (40 mg) were sealed in a glass ampoule under reduced pressure ($10^{-5}$ mbar) and heated to 38 °C for 48 hours. The ampoule was allowed to cool to room temperature before exposing the sample to air. The composite was then stirred in 1.0 M hydrochloric acid for 30 minutes and then washed with deionised water until a neutral pH is seen in the washings, to give the $Co_3O_4$@SWNT composite.

### $RuO_2$@SWNT

$Ru_3(CO)_{12}$ (20 mg, 0.031 mmol) and freshly opened SWNTs (40 mg) were sealed in a glass ampoule under reduced pressure ($10^{-5}$ mbar) and heated to 130 °C for 48 hours. The ampoule was allowed to cool to room temperature before exposing the sample to air. The composite was then washed with acetonitrile (3 x 50 mL), allowed to dry and then heated in air at 150 °C for 24 hours to give the $RuO_2$@SWNT composite.

### $IrO_2$@SWNT

$Ir_4(CO)_{12}$ (20 mg, 0.018 mmol) and freshly opened SWNTs (40 mg) were sealed in a glass ampoule under reduced pressure ($10^{-5}$ mbar) and heated to 160 °C for 48 hours. The ampoule was allowed to cool to room temperature before exposing the sample to air, and then heated in air at 170 °C for 24 hours. The composite was then stirred in concentrated hydrochloric acid for 30 minutes and then washed with deionised water until a neutral pH is seen in the washings to give the $IrO_2$@SWNT composite.

### $Co_3O_4$/activated carbon

$Co_2(CO)_8$ (20 mg, 0.058 mmol) and activated carbon (40 mg) were sealed in a glass ampoule under reduced pressure ($10^{-5}$ mbar) and heated to 38 °C for 48 hours. The ampoule was allowed to cool to room temperature before exposing the sample to air to give the $Co_3O_4$/activated carbon ($Co_3O_4$/AC) composite.

### $RuO_2$/activated carbon

$Ru_3(CO)_{12}$ (20 mg, 0.031 mmol) and activated carbon (40 mg) were sealed in a glass ampoule under reduced pressure ($10^{-5}$ mbar) and heated to 130 °C for 48 hours. The ampoule was allowed to cool to room temperature before exposing the sample to air. The composite was then heated in air at 150 °C for 24 hours to give the $RuO_2$/activated carbon ($RuO_2$/AC) composite.

### $IrO_2$/activated carbon

$Ir_4(CO)_{12}$ (20 mg, 0.018 mmol) and activated carbon (40 mg) were sealed in a glass ampoule under reduced pressure ($10^{-5}$ mbar) and heated to 160 °C for 48 hours. The ampoule was allowed to cool to room temperature before exposing the sample to air, and then heated in air at 170 °C for 24 hours to give the $IrO_2$/activated carbon ($IrO_2$/AC) composite.

### Nanotube plugging

Plugging experiments were performed on a modified GC WE to allow for direct comparison of plugged and unplugged material. To prepare the plugged material, $MO_x$@SWNT was coated onto the GC RDE as above, and LSVs performed in $N_2$-saturated 0.1 M potassium hydroxide. After the OER activity was recorded, the electrode was taken out of solution and 10 μL of a saturated solution of $C_{60}$ in toluene was then dropped onto the coated electrode surface and allowed to dry. This was then washed with excess toluene to remove surface-bound $C_{60}$, before performing LSVs in $N_2$-saturated 0.1 M potassium hydroxide as before.

### Nanotube tip closing

SWNTs were tip-closed following literature procedures[1,2]. Following the heating and acid washing procedure described above, SWNTs were sealed in a quartz ampoule under reduced pressure ($10^{-5}$) mbar and heated to 1000 °C for three hours. The ampoule was allowed to cool to room temperature before the nanotubes were exposed to the atmosphere to give the closed C-SWNT material. $MO_x$/C-SWNT materials were made following the same procedure as the corresponding $MO_x$@SWNT materials.

### Characterization

Transmission electron microscopy (TEM) micrographs were acquired using a JOEL 2100F TEM field emission gun microscope operated at 200 keV. Aberration corrected transmission electron microscopy (AC-HRTEM) was performed using the dedicated low voltage spherical ($C_S$) and chromatic ($C_C$) aberration corrected SALVE TEM operated at 60 kV (Linck et al. PRL 117 (2016) 076101, www.salve-center.de). All TEM samples were prepared by dispersing in isopropyl alcohol and then spotting on a copper grid mounted with a "lacey" carbon film (Agar Scientific UK) and allowing to dry. TEM images were processed using Gatan DigitalMicrograph Software. EDX spectra were acquired for samples mounted on lacey-carbon-coated copper TEM grids using an Oxford Instruments INCA X-ray microanalysis system. Raman spectroscopy measurements were acquired using a HORIBA LabRAM HR spectrometer, with a laser wavelength of 532 nm operating at a power of $ca.$ 0.3 mW and a 600 lines $mm^{-1}$ diffraction grating. The spectral resolution is better than 1.8 $cm^{-1}$ in this configuration. For the G-band focused Raman spectra (Fig. 1), a grating of 1800 lines $mm^{-1}$ is used, and the spectral resolution is better than 0.43 $cm^{-1}$. The detector was a Synapse CCD detector. The Raman shift was calibrated using the zero-order line and the 520.7 $cm^{-1}$ line from an Si(100) reference sample. Spectra were baseline corrected using a linear fitting model. Data presented in the manuscript were averaged over three measurements. Raman spectroelectrochemistry measurements were performed using a HORIBA XploRA INV spectrometer, with a laser wavelength of 785 nm and a 600 lines $mm^{-1}$ diffraction grating. The spectral resolution is better than 3.1 $cm^{-1}$ in this configuration. The detector was a Sincerity CCD. Details of the calibration and spectra processing followed those used for ex situ Raman spectroscopy measurements. Data presented in the manuscript was measured once. A schematic of the cell configuration can be found in Supplementary Fig. 33. A static glassy carbon working electrode was used (0.07 $cm^2$) with the same counter and reference electrode as outlined in the electrochemical characterisation section below. There was no iR-correction applied. The electrolyte volume was 30 mL. Thermogravimetric analysis (TGA) was performed using a TA Q500 Thermogravimetric Analyzer. All samples were analyzed within a platinum pan and in the presence of air. The parameters for all experiments were: ramp 5 °C $min^{-1}$ from 20 to 1000 °C followed by an isotherm for 10 min at 1000 °C, air flow: 60 mL $min^{-1}$. X-ray photoelectron spectroscopy (XPS) was carried out using a Kratos AXIS DLD instrument equipped with an Al $K_\alpha$ X-ray source (1486.6 eV). Data presented in the manuscript was taken from

a single measurement, with a pixel resolution of 0.1 eV. A linear background correction was applied to all spectra prior to analysis. All spectra were charge corrected to the silica peak at 95.5 eV[3].

## Electrochemical characterisation

10 mg of electrocatalyst (MO$_x$@SWNT, MO$_x$/AC, MO$_x$/C-SWNT, MO$_x$) was suspended in 900 μL of DMF and 100 μL of 5% Nafion (in aliphatic alcohols). A 10 μL drop of this ink was dropcast onto a 5 mm glassy carbon rotating disc electrode. This gave a total mass loading of 0.5 mg cm$^{-2}$ on the electrode. Oxygen evolution activity was measured in oxygen-saturated 0.1 mol dm$^{-3}$ potassium hydroxide at 1600 rpm. A Hg/HgO (0.1 M potassium hydroxide, IJ Cambria) reference electrode and graphite rod counter electrode were used. The potential of the Hg/HgO (0.1 M potassium hydroxide, pH 13) reference electrode was measured against either a homemade RHE or a Gaskatel Hydroflex. All potentials are given versus RHE and were converted post-measurement using the correction method described. A potentiostat (CH Instruments, CHI 760c) was used to control the potential. All voltammograms were iR-corrected using CHI software for resistances of approximately 5 – 10 Ω. Linear sweep voltammograms were measured at 5 mV s$^{-1}$ in O$_2$ or N$_2$-saturated electrolyte at 1600 rpm. Stability tests were carried out by repeatedly cycling between 0.9 and 1.7 V vs RHE at 100 mV s$^{-1}$ for 2000 cycles in N$_2$-sparged electrolyte at 1600 rpm. All voltammetry was recorded in a temperature-controlled water bath at 25 °C with an electrolyte volume of 30 mL. Reported electrochemical values were obtained from single experiments, but are representative of reproducible data.

## Computational details

We performed first principles spin-polarized density functional theory (DFT) calculations using the mixed Gaussian and plane wave basis as implemented in the CP2K/QUICKSTEP package version 2023.2[4,5]. The exchange-correlation energy is calculated using the generalized gradient approximation (GGA) using the Perdew–Burke–Ernzerhof (PBE) functional[6], norm-conserving Goedecker–Teter–Hutter (GTH) pseudopotentials[7,8] were adopted and DZVP-MOLOPT-SR-GTH basis sets[9]. A 400 Ry plane wave cutoff was used in all performed calculations. The chemical structures of the free-standing SWNT (11,11) and the encapsulation complex were fully optimized using the limited memory Broyden-Fletcher-Goldfarb-Shanno (L-BFGS) algorithm. Periodic boundary conditions were applied in all the calculations to avoid termination effects on the SWNT and in order to avoid interactions with periodic images in the non-periodic directions we added a vacuum layer of 20 Å. We used SWNTs of ~50 Å length and we placed the MO$_x$ sections in the middle of the SWNT so there is ~18 Å between molecules in the periodic system to avoid interactions between them. Dispersion corrections to consider van der Waals interactions were implemented by applying semi-empirical Grimme-D3 corrections[10]. In order to quantify this electronic density transfer in the encapsulation complexes, we performed both Mulliken and Bader charge analysis to obtain an integer number of transferred electrons. First, we tested the equivalence of both methods and we obtained similar results. Hence, all the results shown were obtained using Mulliken charges analysis due to the direct implementation of this method on the CP2K/QUICKSTEP package.

## Data availability

The authors declare that the data supporting the findings of this study are available within the paper and its supplementary information file.

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

## Acknowledgements

A.N.K. thanks the EPSRC (Established Career Fellowship EP/R024790/1 and MASI Programme Grant EP/V000055/1). D.A.W., L.R.J., and G.N.N. thank the Faraday Institution LiSTAR project (EP/S003053/1, FIRG014) and the University of Nottingham's Propulsion Futures Beacon of Excellence. D.L.-A. and J.J.B. thank the European Union (ERC-2021-StG-101042680 2D-SMARTiES) and the Generalitat Valenciana (grant CIDEXG/2023/1). W.J.V.T. thanks the EPSRC for a 2023 Doctoral Prize (EP/W524402/1). U.K. and J.B. thank the German Research Foundation (DFG) forthe financial support grant 364549901 SFB TRR 234 "Cata-Light" (sub-project Z2). The authors also thank the Nottingham Nanoscale and Microscale Research Centre (nmRC) for providing access to instrumentation (EP/S017739/1) and NanoPrime for in-situ electrochemical Raman access (EP/R025282/1).

## Author contributions

All authors contributed to the conception and design of the study. W. J. V. T., M. B. and J. W. J performed the experiments. D. L.-A. performed the ab initio calculations under the supervision of J. J. B. Raman spectroscopy was conducted in collaboration with G. A. R. AC-TEM analysis was performed by J. B. with U. K. All authors contributed to manuscript writing. D. A. W., L. R. J., A. N. K. and G. N. N. supervised the project.

## Competing interests

The authors declare no competing interests.
