## [Transparent Peer Review file · Nature Communications]

The Role of Carbon Catalyst Coatings in the Electrochemical Water Splitting Reaction

Corresponding Author: Professor Graham Newton

Version 0:

Reviewer comments:

Reviewer #1

(Remarks to the Author)

In this work, a simple and clever method was demonstrated to demonstrate that the active sites of carbon coated metal materials exist in carbon, and to achieve metal activation of carbon active sites through charge transfer mechanisms. The authors demonstrate this result using model catalysts consisting of <2 nm metal-oxide electrocatalyst nanoparticles encapsulated within single-walled carbon nanotubes. There are some shortcomings and insufficient data in this work. The manuscript deserves publication in Nature communication after some major revision.

My comments are as follows:

1. There are many formatting issues in the paper figures, such as the missing annotation of the vertical axis in Figure 1e. The size of the annotation (ABCD) in the figure is not uniform.
2. The authors can perform some elemental mapping analysis in TEM characterization to better explain the distribution of metal oxides in materials.
3. The synthesis process is under reduced pressure (10-5 mbar). The authors need to provide a detailed description of their impact on synthesis.
4. The authors use C60 to isolate metal active sites, but C60 may improve the overall conductivity of the material and enhance its performance. Can the authors design a KSCN poisoning experiment to replace the effect of C60?
5. Can the author describe more about the interaction between metals and carbon support in this material system? And some references should be added in this work to learn more about interaction between metals and carbon support (Please check <https://doi.org/10.1021/acs.chemmater.2c01738>; <https://doi.org/10.1016/j.apcatb.2019.118080> and <https://doi.org/10.1002/idm2.12059>.)
6. Carbon-coatings are proposed to improve electrocatalyst stability and limit active materials' leaching. The authors need characterization after material stability experiments to prove (such as XPS, ICP, etc.)
7. "This charge transfer results in a depletion of electron density on the carbon surface, facilitating the binding of -OH, the likely RDS indicated by electrochemical analysis." The author proposes that the carbon surface enhances its activity by promoting the adsorption of OH-, but this viewpoint is not very convincing. The determining step of OER is generally not the adsorption of hydroxyl groups in the first step. Merely improving the OER activity of the carbon surface through the adsorption effect of hydroxyl groups is not convincing enough. It is suggested that the authors can also deeply analyze and calculate other OER steps in the model.

Reviewer #2

(Remarks to the Author)

In this manuscript, the authors report the synthesis of catalyst consisting of metal-oxide nanoparticle encapsulated within single-walled carbon nanotube. The novelty of this work is limited and the OER performance quite poor. There are large

numbers of reported electrocatalysts about carbon-coated metal/metal oxide nanoparticles and they show excellent OER activity. Besides, several major claims suggested by the authors lack sufficient experimental evidences, particularly regarding the mechanism of charge transfer, the role of carbon coating and the origin of OER activity. Overall, I cannot support its publication in Nature Communications. Some detailed comments are provided as below.

1. The evidence of charge transfer is not convincing. The authors confirmed the charge transfer only by Raman, and the results were not solid. More characterization methods should be carried out, for example, X-ray absorption spectra, ultraviolet photoelectron spectroscopy, etc.
2. XPS only includes the carbon-coated samples. The control samples without carbon coatings (Co₃O₄, RuO₂, IrO₂) should also be characterized by XPS, which is the important evidence of charge transfer.
3. The metal contents in catalysts are suggested to be analyzed by ICP.
4. What is the influence of length and width of carbon nanotube on catalytic behaviors? Have the sizes of nanoparticles been optimized?
5. Electrochemical data of metal oxide nanoparticles without carbon coatings, commercial RuO₂ and IrO₂ should be included.
6. Electrochemical impedance spectroscopy of catalyst should be measured to obtain the charge transfer resistance.
7. The authors only studied the single-walled carbon nanotube. The numbers of layer play vital role for the catalytic activity and stability, which should be investigated.
8. The intrinsic activity of the catalyst is suggested to be tested and evaluated.
9. Chronopotentiometry curve should be measured to confirm the stability of catalyst.
10. The activity origin of the carbon-coated catalyst is elusive and the carbon coating itself as the active site is questionable. The oxidation peaks of Co species indicated the participation to OER, so how to exclude the contribution for OER activity. Moreover, the structural characterization of post-stability catalyst should be performed to further reveal the active sites.
11. Bader charge and density of states should be provided by density functional theory calculations.
12. The language and layout of figures of this manuscript need to be improved.
13. The authors should carefully check the main text to ensure no errors. For example, "the first step in the OER and the RDS (as determined by electrochemical analysis) and may be key to the activity of carbon coated electrocatalysts", "-OH", and Refs. 30, 44, 53.

Reviewer #3

(Remarks to the Author)

This manuscript reports a development of model electrocatalyst with carbon encapsulation to identify catalytic active site in alkaline oxygen evolution reaction (OER). The model catalyst was designed by encapsulating metal oxides (i.e., Ir, Ru, and Co oxides) within single-walled carbon nanotube (SWNT) and capping with fullerene at the end. The synthesized model catalysts showed little response of redox behavior of oxide catalyst, implying successful encapsulation of metal oxide surface by the carbon layer. The prepared catalysts revealed comparable OER activities to the literature without carbon layers and superior stability, based on which the authors argue that carbon surface on metal oxide as major OER active site. Additional in situ Raman spectroscopic analysis and DFT calculation indicate that charge transfer at carbon-metal oxide junction leads to more positively polarized carbon surface during anodic OER condition which can provide beneficial active site for hydroxide adsorption as rate-determining step for OER. The reviewer evaluates this model study approach is well-designed for identification of OER active site. However, several points need to be addressed more thoroughly with major revisions to enhance clarity and improve the quality of manuscript to be suitable for publication in Nature Communications.

1. In this manuscript, the authors argue electronic structure modification of carbon surface by encapsulated metal oxide as alkaline OER active site. Although both in situ Raman and DFT results clearly show positively charged carbon surface, the reviewer wonders if the modified electronic structure of carbon layer is comparable to the electronic structure of bare metal oxide surfaces. If not, charge transfer will be dependent on work function difference between carbon layer and metal oxide surface, which possibly implies different effect of carbon layer depending on metal oxide identities. So more general discussion on carbon-metal oxide charge transfer is required in terms of charge transfer.
2. The SWNT-encapsulated metal oxide catalysts are developed with unique approach. However, its activity comparison to the non-encapsulated (or bare) catalysts are limited in this manuscript and only done by comparison with literature. If the authors first plug fullerene to SWNT and then deposit metal oxide precursor for catalyst preparation, is it possible to prepare metal oxide catalysts selectively outside of the SWNT? If possible, these catalyst group will provide more clear comparison between OER process on bare MO_x surface and carbon-coated MO_x surface.
3. In Figure 3f inset and S15, the authors demonstrated suppressed anodic metal oxidation in Co₃O₄ and RuO₂ when encapsulated by SWNT. This result is reasonable in case of complete encapsulation by carbon layer due to the limited access of electrolyte to metal oxide through the carbon layer. However, in this case, it is anticipated that valence state of metal oxide will be different between bare metal oxide and the encapsulated metal oxide at the same anodic potential (i.e., OER operating condition). Have the authors confirmed this with additional analysis such as X-ray absorption spectroscopy? Also, the reviewer wonders how modified metal valence state by the carbon layer can affect OER electrocatalysis compared to the bare metal oxide.
4. In ex situ Raman analysis (Figure S12), a blueshift of the G-band peak does not happen in Co₃O₄@SWNT in contrary to RuO₂@SWNT and IrO₂@SWNT. The author indicates that the charge transfer is less significant for Co₃O₄@SWNT. However, in situ Raman shift of G band position and DFT results are provided as a proof of modified electronic structure on the carbon surface for Co₃O₄@SWNT at the same time. This discrepancy should be further discussed in detail to fully support the authors' argument.
5. In Figure 3, OER activities are compared between before and after fullerene plugging in MO_x@CWNT. While RuO₂ and IrO₂ catalysts clearly show negligible change through the fullerene plugging, Co₃O₄@SWNT catalyst shows a distinct

increase in OER overpotential in overall current range. Does this activity loss come from loss of Co₃O₄ surface at the end of Co₃O₄@CWNT structure? If it is, does it indicate much higher TOF of Co₃O₄ surface exposed at the end of Co₃O₄@SWNT structure compared to the Co₃O₄ surface encapsulated by CWNT? Moreover, why this feature occurs only in Co₃O₄@CWNT compared to RuO₂@SWNT and IrO₂@SWNT? Plus, why does the plugged Co₃O₄@SWNT show decreased OER activity at low current density (below ~2 mA cm⁻²) compared to the bare SWNT?

6. In addition to the 5th comment, it is necessary to show OER activity of fullerene-plugged SWNT (C₆₀@SWNT) without metal oxide as comparison to confirm negligible role of fullerene in OER activity (likely in Supporting Information).

7. Have the authors applied this approach to acidic OER electrocatalysis (such as IrO₂@SWNT)? The reviewer wonders if there is any change in electrocatalytic mechanism between adsorbate evolution mechanism and lattice oxygen evolution mechanism.

Version 1:

Reviewer comments:

Reviewer #1

(Remarks to the Author)

The author's additional comments have resolved most of our doubts. This is a high-quality and novel paper, and we recommend its publication in nature communications.

Reviewer #2

(Remarks to the Author)

The authors have properly addressed my concerns and I could now support the manuscript for publication.

Reviewer #3

(Remarks to the Author)

This manuscript reports model study of carbon-encapsulated metal oxide nanocrystals as model catalyst for alkaline oxygen evolution reaction (OER) with several revision points added to the previous version. The reviewer evaluates that all comments addressed by the authors make clear explanation to the reviewer questions and are properly updated to the revised manuscript. Therefore, the reviewer recommend the revised manuscript for publication in Nature Communications.

Response to Reviewer 1

In this work, a simple and clever method was demonstrated to demonstrate that the active sites of carbon coated metal materials exist in carbon, and to achieve metal activation of carbon active sites through charge transfer mechanisms. The authors demonstrate this result using model catalysts consisting of <2 nm metal-oxide electrocatalyst nanoparticles encapsulated within single-walled carbon nanotubes. There are some shortcomings and insufficient data in this work. The manuscript deserves publication in Nature communication after some major revision.

We would like to thank the reviewer for taking the time to review our manuscript and for their positive assessment. Below we have answered all major criticisms.

1. There are many formatting issues in the paper figures, such as the missing annotation of the vertical axis in Figure 1e. The size of the annotation (ABCD) in the figure is not uniform.

We thank the reviewer for their keen observations, and we have endeavoured to fix all formatting issues. The missing annotation has been amended, and all figure annotations are uniform.

2. The authors can perform some elemental mapping analysis in TEM characterization to better explain the distribution of metal oxides in materials.

EDX-STEM mapping has now been performed on all MO_x @SWNT materials to better understand the distribution of metal oxides, as well as confirming the absence of any other metal impurities (Figure R1). EDS STEM mapping revealed a uniform distribution of the metal species across a large area of SWNT bundles, highlighting the efficacy of the metal oxide filling method. These data are now presented as Figures S9 and discussed on page 4 of the main manuscript.

“Energy-dispersive X-ray spectroscopy (EDX) mapping was performed on these materials during TEM imaging, which revealed characteristic Ir, Ru and Co peaks for IrO_2 @SWNT, RuO_2 @SWNT and Co_3O_4 @SWNT, respectively (Fig. S9). This EDX mapping showed a uniform distribution of carbon, oxygen and the corresponding metal for each of the MO_x @SWNT materials, confirming that the encapsulated metals were present throughout the sample.”

Figure R1 | Scanning transmission electron microscopy (STEM) energy dispersive X-ray (EDX) maps and corresponding spectra of $\text{Co}_3\text{O}_4@\text{SWNT}$, $\text{IrO}_2@\text{SWNT}$ and $\text{RuO}_2@\text{SWNT}$. Small peaks at 2.1, 5.4 and 6.4 eV are due to gold, chromium and iron arising from the sample holder.

3. The synthesis process is under reduced pressure (10⁻⁵ mbar). The authors need to provide a detailed description of their impact on synthesis.

We thank the reviewer for highlighting our lack of clarity here. Reduced pressure is used to ensure ease of sublimation of the organometallic species as precursors to metal oxides. This method is a common synthetic procedure for filling SWNTs.¹⁻³ We have verified that this procedure does not affect SWNT quality by observing a very low defect density (from D-band intensity in Raman spectroscopy) before and after the filling procedure (Figure S14). We now discuss this on page 5 of the main manuscript and present the data in a modified version of Table S1.

“All samples measured had low levels of structural defects, with calculated densities <12 defects μm⁻¹ (Table S1), demonstrating that the experimental conditions during encapsulation of metal oxides led to only small increases in the number of defect sites, and that the encapsulated metal oxide sites were effectively separated from the electrolyte by the SWNT walls.”

4. The authors use C₆₀ to isolate metal active sites, but C₆₀ may improve the overall conductivity of the material and enhance its performance. Can the authors design a KSCN poisoning experiment to replace the effect of C₆₀?

We thank the reviewer for raising the question of potential performance benefits induced by the interaction between SWNTs and C₆₀. For clarity, we have repeated the plugging experiments with unfilled SWNTs and found almost little change in electrocatalytic activity before and after plugging (Figure R2). We present this data in Figure S23 and discuss the implications on page 8 of the main manuscript.

“Finally, to test whether fullerene encapsulation has an inherent effect on the OER activity of the carbon shell, we plugged pristine SWNTs with fullerenes. No significant change in OER activity was observed after plugging (Fig. S23).”

Figure R2 | RDE-LSVs of SWNT and C₆₀-SWNT, recorded in N₂ saturated 0.1 M potassium hydroxide at 5 mV s⁻¹ and 1600 rpm.

We performed KSCN poisoning experiments on the encapsulated RuO₂@SWNT and IrO₂@SWNT systems, and compared the results to those obtained using MO_x/AC materials, where the metal is exposed to the electrolyte solution. In each case, we first measured the catalytic activity, then added 0.1 M KSCN and waited 30 minutes before measuring activity again, following literature precedent (Figure R3).⁴ We found that while MO_x/AC materials exhibited a large drop in OER activity after addition of KSCN, the encapsulated materials exhibited almost no change in OER activity. This data confirms that the metal oxides within the MO_x@SWNT materials are not exposed to the electrolyte, as KSCN addition does not lead to a drop in performance. Moreover, it suggests that the carbon surface may be resistant to some poisoning effects that trouble metal oxide catalysts. We note that Co₃O₄@SWNT and Co₃O₄/AC are not included as we observed an unexpected increase in OER activity, attributable to KSCN oxidation, as observed for some metal/metal oxide systems in the literature.^{5,6} We present this data in Figure S18 and it is discussed on page 8 of the main manuscript.

“To further test whether the carbon surface, and not the encapsulated metal oxide surface, was the active site, a series of OER measurements were run after addition of KSCN to the electrolyte; KSCN binds to metal-oxide surfaces, limiting their electrocatalytic activity. No change in the activity of the MO_x@SWNT materials was apparent, while KSCN caused a drastic decrease in the activity of the corresponding MO_x/AC materials, demonstrating further that the carbon surface was most likely the active site in the MO_x@SWNT materials (Fig. S18).”

Figure R3 | KSCN poisoning experiments of RuO₂@SWNT, RuO₂/AC, IrO₂@SWNT and IrO₂/AC (a-d) respectively. LSVs were performed in 1.0 M KOH using a scan rate of 5 mV/s, before addition 0.1 M KSCN. The modified working electrode was allowed to rotate in 1.0 M KOH and 0.1 M KSCN for 30 minutes before another LSV was performed.

5. Can the author describe more about the interaction between metals and carbon support in this material system? And some references should be added in this work to learn more about interaction between metals and carbon support (Please check <https://doi.org/10.1021/acs.chemmater.2c01738>; <https://doi.org/10.1016/j.apcatb.2019.118080> and <https://doi.org/10.1002/idm2.12059>.)

We thank the reviewer for bringing our attention to these important references. These have been added to the manuscript in the introduction on page 2 and 3 of the main manuscript, along with some more discussion on the current understanding of electronic interactions between metals and carbon supports. We have also gone into more detail on carbon – metal oxide interactions in the discussion of our DFT calculations on page 10 of the main manuscript.

“A systematic charge transfer from the C π-electron system to the MO_x can be observed in all cases and the transfer was predominantly to the outer oxygen atoms of the encapsulated metal oxides. As the terminal O atoms in the transition metal oxide are highly electronegative, they attract electron density

from the delocalized π system on the inner part of the SWNT. The charge transfer was not necessarily homogenous, with the largest occurring on MO_x oxygen atoms closest to the inner part of the SWNT (Fig. S28 and 29). This demonstrates that the electronic structure and morphology of the encapsulated material affected charge transfer.”

6. Carbon-coatings are proposed to improve electrocatalyst stability and limit active materials' leaching. The authors need characterization after material stability experiments to prove (such as XPS, ICP, etc.)

It is important to note that we are not proposing that our carbon coatings necessarily lead to large improvements in stability or that they are practical solutions, and rather we are using them to understand the mechanism of OER on carbon-coated materials. The sentence highlighted by the reviewer is a reference to assumptions in the field by other authors. We have edited our manuscript on pages 3 and 8, as well as adding in reference [7], to clarify that we are working with a model system and that we are referring to carbon coatings as seen in the literature in general.

“It is important to note that we do not expect that these “ $MO_x@SWNT$ ” materials are likely to become practical electrocatalysts for devices, rather our work demonstrates that modulation of the electronic structure of carbon can drastically affect its electrocatalytic behaviour. ”

“Therefore, encapsulation of metal oxides within SWNTs appears to protect the confined species in the $MO_x@SWNT$ materials.”

As requested, we have performed some post-cycling analysis to better understand how our materials change over time. ICP was conducted on the electrolyte solution after cycling for 2000 cycles and we found no (<ppm level) leached metal species from any of the $MO_x@SWNT$ materials. A comment has been added to page 6 of the manuscript.

“... while inductively Coupled Plasma (ICP) analysis of all electrolytes showed no detectable leaching of metal ions after 2000 cycles.”

The Raman spectra of all $MO_x@SWNT$ materials and SWNT were acquired before and after 2000 cycles. A small increase in the number of defects is observed, indicated by the relatively small increase in $I_D:I_G$ ratios (Figure R4), suggesting that there is some carbon degradation over time. These data are presented in Figure S14 and Table S1 are discussed in the main manuscript on page 6.

“Raman spectroscopy of the materials before and after cycling (Fig. S14) revealed less than 20 defects μm^{-1} in the SWNTs (Table S1)”

Figure R4 | (a-d) Raman spectra of Co₃O₄@SWNT, RuO₂@SWNT, IrO₂@SWNT and SWNT respectively, before and after cycling 2000 times in 0.1 M KOH. Spectra highlight the D band, with intensities normalised to the G band.

Table S1 | Mass loadings, taken from TGA (Fig. S1), and defect densities taken from 532 nm Raman spectra, of the encapsulated materials and SWNT control. The defect density of each material is calculated using the integrated Raman D/G ratios.¹¹ Atomic percent is calculated using mass loading taken from TGA.

Catalyst	Mass loading MO _x / wt. %	Atomic percent of M / %	Defect density / μm ⁻¹	2000 cycle defect density / μm ⁻¹
IrO ₂ @SWNT	15.8	0.99	8.4 ± 0.2	11.8 ± 0.3
RuO ₂ @SWNT	12.5	1.27	7.5 ± 0.2	10.8 ± 0.3
Co ₃ O ₄ @SWNT	6.1	0.96	11.2 ± 0.3	19.3 ± 0.5
SWNT	N/A	N/A	5.0 ± 0.1	14.3 ± 0.4

7. "This charge transfer results in a depletion of electron density on the carbon surface, facilitating the binding of -OH, the likely RDS indicated by electrochemical analysis." The author proposes that the carbon surface enhances its activity by promoting the adsorption of OH⁻, but this viewpoint is not very convincing. The determining step of OER is generally not the adsorption of hydroxyl groups in the first step. Merely improving the OER activity of the carbon surface through the adsorption effect of hydroxyl groups is not convincing enough. It is suggested that the authors can also deeply analyze and calculate other OER steps in the model.

We agree that for typical metal oxide systems, the rate-determining step of the OER is often not the adsorption of hydroxyl groups, as demonstrated by their low Tafel slopes.⁸⁻¹⁰ However, in these materials, as the active site is carbon, we believe the binding of ⁻OH is the RDS. This is in line with Tafel slope values¹⁰ and literature precedence for the RDS on SWNTs in the reaction conditions used in this study.¹¹ We now highlight these differences in RDS with the relevant references on page 6 of the main manuscript.

"Tafel analysis of the RDE voltammetry data (Fig. 2b and Table S2) yielded OER Tafel slopes of about 100 mV dec⁻¹ for Co₃O₄@SWNT, IrO₂@SWNT and empty SWNTs, indicating that the rate-determining step (RDS) was the same in all three systems. The values obtained are close to those obtained when binding of ⁻OH ions to the active site is the RDS,⁷⁶ as reported for SWNTs in alkaline conditions.⁷⁷"

Response to Reviewer 2

In this manuscript, the authors report the synthesis of catalyst consisting of metal-oxide nanoparticle encapsulated within single-walled carbon nanotube. The novelty of this work is limited and the OER performance quite poor. There are large numbers of reported electrocatalysts about carbon-coated metal/metal oxide nanoparticles and they show excellent OER activity. Besides, several major claims suggested by the authors lack sufficient experimental evidences, particularly regarding the mechanism of charge transfer, the role of carbon coating and the origin of OER activity. Overall, I cannot support its publication in Nature Communications. Some detailed comments are provided as below.

We are grateful to the reviewer for taking the time to review our manuscript and for highlighting some deficiencies in our work. They rightly highlight that proving the charge transfer effect is a key point in understanding the mechanism of carbon-coated electrocatalysts. We hope the new data, given below, provides unambiguous evidence for the charge depletion of carbon within our materials and proof that the electrolysis can only be occurring at the carbon surface.

1. The evidence of charge transfer is not convincing. The authors confirmed the charge transfer only by Raman, and the results were not solid. More characterization methods should be carried out, for example, X-ray absorption spectra, ultraviolet photoelectron spectroscopy, etc.

In the case of the Raman spectroscopy, we have repeated the results with a higher spectral resolution instrument. Specifically, we used a higher grating of 1800 mm^{-1} , giving spectral resolution of 0.43 cm^{-1} as compared to 1.55 cm^{-1} previously. Using these new parameters, a clear trend in the position of the G band can be seen in each material (Figure R5), which follows the stepwise magnitude of charge transfer shifts we have calculated computationally (3.49, 2.83 and 1.04 electrons transferred per modelled MO_x unit shown in Figure 4c, S28 and 29 for $\text{RuO}_2@SWNT$, $\text{IrO}_2@SWNT$ and $\text{Co}_3\text{O}_4@SWNT$ respectively, see discussion on page 10 of the main manuscript). This has been presented in the main text in Figure 1f and discussed on page 5 of the main manuscript.

“The G-band positions in the spectra of all MO_x were blueshifted relative to that of unfilled SWNT (Fig. 1f), with $\text{RuO}_2@SWNT$ displaying the largest shift, followed by $\text{IrO}_2@SWNT$ and then $\text{Co}_3\text{O}_4@SWNT$. These shifts were in line with XPS analysis and further evidence of charge transfer from the SWNT to the encapsulated metal oxides.⁶⁶”

Figure R5 | Raman spectra showing the G band of RuO₂@SWNT, IrO₂@SWNT, Co₃O₄@SWNT and SWNT (a) and highlighted G band maxima (b).

To further investigate this charge transfer from the carbon to the metal oxide in each of these materials we have revisited XPS as a tool to probe the electronic structure of carbon. Monitoring the shift in the carbon peak can cause difficulties in XPS, as the adventitious carbon peak is often used as an internal reference for charge correction. Obviously, correcting to the carbon peak means that we cannot accurately compare changes in the carbon electronic structure. To combat this, we loaded the samples onto silicon wafers, which have been extensively studied with XPS.¹² When taking XPS measurements, we ensured that the X-ray beam was over an area containing both our catalyst and bare Si surface. This meant that we could use the Si 2p peak at 99.5 eV¹² as an internal reference for charge correction. Using this internal reference; when comparing the C 1s peaks, a shift to higher binding energies (consistent with a reduction in electron density) is observed for all MO_x@SWNT materials compared to bare SWNTs (Figure R6). Again, this agrees with the trend observed in both the G-band shift in Raman spectroscopy and the computational results, with RuO₂@SWNT displaying the largest shift in binding energy resulting from the largest charge transfer. This data unambiguously shows that the encapsulated metal oxide results in depletion of electron density from carbon shell, consistent with our theory. This has been added to Figure 1e in the main text, along with discussion on page 4 of the main manuscript.

“XPS of the MO_x@SWNT materials was performed to elucidate the properties of the carbon and the chemical states of the confined MO_x (Fig. 1e and S10 and 11). To accurately compare the position of the carbon 1s peak, the samples were loaded onto silicon wafers, and the Si 2p peak at 99.5 eV⁶⁴ was used as an internal reference to correct for charge accumulation (Fig. S10). The carbon peaks of all MO_x@SWNT were shifted to a higher binding energy compared to those of unfilled SWNTs, with RuO₂ displaying the largest shift. This shift suggests that electron density on the carbon surface decreased due to charge transfer to the encapsulated metal oxide, which is consistent with reports of similar materials.⁶³”

Figure R6 | X-ray photoelectron spectra showing the C 1s region of RuO₂@SWNT, IrO₂@SWNT, Co₃O₄@SWNT and SWNT. Inset shows the highlighted peak maxima. Wide-scan spectra of each material can be found in the SI (Fig. S10).

2. XPS only includes the carbon-coated samples. The control samples without carbon coatings (Co₃O₄, RuO₂, IrO₂) should also be characterized by XPS, which is the important evidence of charge transfer.

We have performed XPS on the MO_x@SWNT and MO_x/AC materials. While the revised XPS data showed clear evidence of charge transfer in the C 1s peak (Figure R6), comparison of the metal peaks is more difficult (Figure R7). This is due to a variety of factors affecting the shape of the peaks; the fact that carbon in a single atomic layer encapsulates the metal oxides, and that the metal oxides are multi-atom particles. This data is now included as Figure S11 and discussed in Supplementary Note 2 on page 13 of the supplementary information.

Figure R7 | XP spectra of a) Co 2p region of Co₃O₄@SWNT (top) and Co₃O₄/AC (bottom), b) Ru 3p region of RuO₂@SWNT (top) and RuO₂/AC (bottom) and c) Ir 4f region of IrO₂@SWNT (top) and IrO₂/AC (bottom).

3. The metal contents in catalysts are suggested to be analyzed by ICP.

While ICP is a useful tool to precisely measure elemental composition for many materials, complete digestion of the carbon coating presents a challenge for X@SWNT materials.¹³ Without complete digestion, the carbon shell around the encapsulated metal means that the metal ions cannot be detected, resulting in a much lower than predicted weight loading, while also giving different results based on digestion methods or metals of interest.¹⁴

As an alternative to ICP, we have provided new XPS and EDX data (Figure R8 and R9), confirming that no other metals or elements are present in the material. As a result, we are confident that the remaining material after heating to 1000 °C during TGA analysis is the metal oxides only and provides an accurate mass loading. The new EDX data is presented in Figure S9 and S10, and a discussion of this process has been added to page 4 of the main manuscript.

“Energy-dispersive X-ray spectroscopy (EDX) mapping was performed on these materials during TEM imaging, which revealed characteristic Ir, Ru and Co peaks for IrO₂@SWNT, RuO₂@SWNT and Co₃O₄@SWNT respectively (Fig. S9). EDX mapping showed a uniform distribution of carbon, oxygen and the corresponding metal for each of the MO_x@SWNT materials, confirming that the encapsulated metals were present throughout the sample. These data are further supported by XPS analysis confirming the expected elemental composition of each catalyst (Figure S10).”

Figure R8 | Scanning transmission electron microscopy (STEM) energy dispersive X-ray (EDX) maps and corresponding spectra of $\text{Co}_3\text{O}_4@\text{SWNT}$, $\text{IrO}_2@\text{SWNT}$ and $\text{RuO}_2@\text{SWNT}$. Small peaks at 2.1, 5.4 and 6.4 eV are due to gold, chromium and iron arising from the sample holder.

Figure R9 | XPS spectra of a) Co_3O_4 @SWNT, b) RuO_2 @SWNT and c) IrO_2 @SWNT. Inset is the high-resolution Si 2p region taken from each sample, originating from the Si wafer support, which is used here as an internal reference for charge correction.

4. What is the influence of length and width of carbon nanotube on catalytic behaviors? Have the sizes of nanoparticles been optimized?

We thank the reviewer for raising this important point. The size and shape of nanotubes used in this study is very important. Unlike previous studies of X@CNT, these single walled carbon nanotubes are very narrow (~1.5 nm diameter), limiting access of the electrolyte into the inner pores and thus to the surface of the metal oxides. This, along with the potential to block the internal channels with fullerenes is key in this study as it allows us to eliminate electrolyte access pathways and focus entirely on the chemistry of the carbon. These single-walled carbon nanotubes in particular were chosen due to the very low defect density (<12 μm) and uniformity of diameter and length. While other nanotubes can be purchased with different diameters and lengths, we found experimentally very poor diameter uniformity and high defect densities in these other nanotubes. The decision was made to focus on these SWNTs, as a high uniformity is key in relating catalytic performance to a mechanistic understanding.

As for the size of the encapsulated metal oxide nanoparticles, the nanotube itself acts as a template, controlling nanoparticle dimensions to widths of < 1 nm. The length of the nanoparticles was measured

using TEM and ranged from ~2 nm to ~10 nm. However, the absolute size and uniformity of encapsulated metal oxides is difficult to monitor with TEM due to electron beam-induced dynamics, as previously seen in investigations of MO_x@SWNT materials.¹⁵

We have added in some discussion on the properties of the SWNTs used in this study, why they were selected, and their impact of metal oxide particle dimensions, to the main manuscript on page 3.

“The SWNTs have low defect densities and small diameter distribution, which is essential when using bulk electrocatalytic activity to rationalise a mechanistic pathway and to limit access of the electrolyte into the inner pores and thus to the surface of the metal oxides.”

5. Electrochemical data of metal oxide nanoparticles without carbon coatings, commercial RuO₂ and IrO₂ should be included.

We thank the reviewer for their suggestion. It is important to note that we do not claim that our materials are optimised practical catalysts, but the MO_x@SWNT materials are used here as a model system to understand electrocatalysis at carbon that has been modified by an underlying metal oxide. We have now tried to emphasise this in our revised introduction (page 3).

“It is important to note that we do not expect that these “MO_x@SWNT” materials are likely to become practical electrocatalysts for devices, rather our work demonstrates that modulation of the electronic structure of carbon can drastically affect its electrocatalytic behaviour.”

As stated in our manuscript, these materials display metal oxide-like performance, but at a carbon surface.

As requested, LSVs of the OER at MO_x@SWNT have been compared to metal oxides loaded onto activated carbon (MO_x/AC; experimental details can be found in Supplementary Information) and commercial catalysts (purchased from Sigma Aldrich) and the resulting data are shown in Figure R10. To further investigate encapsulated vs bare metal oxides, we also investigated metal oxides loaded onto tip-closed SWNTs (MO_x/C-SWNT), where metal oxides are only on the nanotube surface and not in the nanotube interior (See the experimental details in the Supplementary Information and our response to Reviewer 3 Question 2 for greater detail and discussion of these materials). As suggested by our initial assessment, the MO_x@SWNT materials display performance similar or better than all standard MO_x catalysts, consistent with our proposal that they display “metal oxide-like performance”.

These data are presented in Figure S15, and a discussion of the data has been added to page 8 of the main manuscript.

“For comparison with our MO_x@SWNT materials, Co₃O₄, RuO₂ and IrO₂ were also deposited onto activated carbons to yield MO_x/AC (see Supplementary Information for details), and OER activities and

stabilities were measured (Fig. 2c-e, S15). The initial activities of the MO_x/AC controls were comparable to those of the MO_x@SWNT analogues,....”

Figure R10 | a-c) LSVs of Co₃O₄@SWNT, IrO₂@SWNT and RuO₂@SWNT respectively, compared to relevant MO_x/AC and commercially available MO_x materials in O₂-saturated aqueous 0.1 M potassium hydroxide solutions.

6. Electrochemical impedance spectroscopy of catalyst should be measured to obtain the charge transfer resistance.

As requested, impedance spectroscopy of the catalysts has been recorded at a constant potential of 0.9 V and 1.5 V vs RHE. The resulting Nyquist plots (Figure R11) display a high frequency semicircle, related to the material, which is seen at all potentials (Figure R11a), and a lower frequency signal which relevant literature suggests to be the charge transfer resistance for the OER. These data are now included as Figure S30 and discussed on page 10 of the main manuscript:

“This demonstrates that the electronic structure and morphology of the encapsulated material affected charge transfer. In addition, differences in the charge-transfer resistance at OER potentials were observed (Fig. S31) and depended on the identity of encapsulated MO_x.”

Figure R11 | Nyquist plots of SWNT, Co₃O₄@SWNT, IrO₂@SWNT and RuO₂@SWNT, taken at a) 0.9 V and b) 1.5 V vs RHE in N₂ saturated 0.1 M KOH, using frequencies between 100 kHz and 0.1 Hz at 1600 rpm.

7. The authors only studied the single-walled carbon nanotube. The numbers of layer play vital role for the catalytic activity and stability, which should be investigated.

We thank the reviewer for the suggestion on considering multiple layers of carbon, we agree that this would serve as an interesting comparison and provide more information as to the degree of charge transfer between the encapsulated metal oxides and carbon layers.

Unfortunately, acquiring carbon nanotubes with > 1 layer but with similar internal diameters and narrow size distribution proved challenging. In particular, some double-walled carbon nanotubes were purchased but when investigated using TEM, we found a wide size distribution, with nanotubes of diameter > 10 nm and multiple concentric layers present (Figure R12). To accurately relate the bulk electrocatalytic properties to the physicochemical properties of our hybrid materials, total confidence in the uniformity of the sample is required, rendering nanotubes with multiples layers unrealistic for this study. We have added some discussion in the main text on page 3 to clarify our choice of carbon nanotubes utilised in this study.

“The SWNTs chosen in our study have a low defect density and small diameter distribution, which is essential when using bulk electrocatalytic activity to rationalise a mechanistic pathway.....”

Figure R12 | a-d) TEM images of DWNTs showing nanotubes of different diameters and wall numbers, as well as defective carbon. TEM images were taken with JOEL 2100F TEM field emission gun microscope operated at 200 keV.

8. The intrinsic activity of the catalyst is suggested to be tested and evaluated.

As discussed above, we do not propose that our materials are a practical solution for OER or that the performance will exceed other known metal oxide catalysts for OER. However, as requested by the reviewer, we have included the intrinsic activity of the catalyst by comparing the mass activity (given that specific activity is hard to determine in this case) in A/g_{MO_x} of the $MO_x@SWNT$ to that of the commercially available catalyst (Figure R13). We see that for all catalysts, the intrinsic activity of the encapsulated metal oxides far exceeds that of the bare metal oxides. In this instance, we do not include the data in the main manuscript as we feel this may make the readers think that our materials could be practical OER catalysts, which is not our intention.

Figure R13 | a-c) LSVs presented as mass activities of Co_3O_4 @SWNT, IrO_2 @SWNT and RuO_2 @SWNT respectively, compared to relevant commercially available MO_x in O_2 -saturated aqueous 0.1 M potassium hydroxide solutions. Mass activity is calculated based on the mass of MO_x drop-cast on the electrode.

9. Chronopotentiometry curve should be measured to confirm the stability of catalyst.

While we agree with the reviewer that chronopotentiometry measurements are useful for evaluating practical catalyst stability, as we note above, we are not proposing that our materials are practical catalysts, but just model systems to probe OER catalysis at carbon surfaces with an underlying metal oxide. As such we believe further studies in this area only misdirect our paper. We have edited the main manuscript on page 3 to clarify the purpose of our paper, the specific insights we hope to reveal and to state that we are not evaluating performance.

“It is important to note that we do not expect that these “ MO_x @SWNT” materials are likely to become practical electrocatalysts for devices, rather our work demonstrates that modulation of the electronic structure of carbon can drastically affect its electrocatalytic behaviour.”

10. The activity origin of the carbon-coated catalyst is elusive and the carbon coating itself as the active site is questionable. The oxidation peaks of Co species indicated the participation to OER, so how to exclude the contribution for OER activity. Moreover, the structural characterization of post-stability catalyst should be performed to further reveal the active sites.

We appreciate the reviewer's concerns on providing further evidence for the carbon surface being the active site and have provided significant new data to further prove our theory. As discussed above, the XPS data in Figure R6 provides proof of the charge depletion of the carbon. In the original manuscript, we showed that fullerene plugging results in loss of the cobalt oxidation process while retaining electrocatalytic activity, confirming that Co is not involved in the OER. To further exclude the metal oxide surface as a potential active site, we have run poisoning experiments as requested by Reviewer 1 (Figure R14), which show that while metal oxide catalysts that bind the poisoning anion display a drop in activity, our encapsulated catalysts are not poisoned, further supporting our proposal that the carbon surface is acting as an active site. This data is included in the revised supporting information as Figure S18 and discussed on page 8 of the main manuscript.

“To further test whether the carbon surface, and not the encapsulated metal oxide surface, was the active site, a series of OER measurements were run after addition of KSCN to the electrolyte; KSCN binds to metal-oxide surfaces, limiting their electrocatalytic activity. No change in the activity of the $MO_x@SWNT$ materials was apparent, while KSCN caused a drastic decrease in the activity of the corresponding MO_x/AC materials, demonstrating further that the carbon surface was most likely the active site in the $MO_x@SWNT$ materials (Fig. S18).”

Figure R14 | KSCN poisoning experiments of RuO₂@SWNT, RuO₂/AC, IrO₂@SWNT and IrO₂/AC (a-d) respectively. LSVs were performed in 1.0 M KOH using a scan rate of 5 mV/s, before addition 0.1 M KSCN. The modified working electrode was allowed to rotate in 1.0 M KOH and 0.1 M KSCN for 30 minutes before another LSV was performed.

Regarding the post-cycling active sites, we expect during cycling that some degradation of the material will occur and that the active site will change, indeed this is confirmed by the change in the Tafel slopes. Therefore, we do not believe analysis after longer cycling periods provides any insight into the active site at carbon for our model system. We discuss the degradation on page 6 of the main manuscript.

“The OER Tafel slopes decreased to 83, 67 and 77 mV dec⁻¹ for Co₃O₄@SWNT, RuO₂@SWNT and IrO₂@SWNT, respectively, after 2000 cycles (Table S2). These changes are likely induced by partial loss of carbon from the SWNT walls leading to exposure of MO_x to the electrolyte or by the introduction of oxygen-containing groups on the surface of the SWNTs during OER, which could lead to a change in local electron density on the carbon surface. This local charge redistribution on the carbon surface can lead to a higher binding energy for ⁻OH.⁴³”

In addition, we performed post cycling analysis to better understand how the materials change over time. ICP was conducted on the electrolyte solution after cycling for 2000 cycles and we found no (<ppm level) leached metal species from any of the MO_x@SWNT materials. The Raman spectra of all MO_x@SWNT materials and SWNT were acquired before and after 2000 cycles. A small increase in the number of defects is observed, indicated by the I_D:I_G ratios (Figure R15), suggesting that we observe some carbon degradation over time. This data is presented in Figure S14 and discussed in the main manuscript on page 6.

“Raman spectroscopy of the materials before and after cycling (Fig. S14) revealed less than 20 defects μm⁻¹ in the SWNTs (Table S1), while inductively Coupled Plasma (ICP) analysis of all electrolytes showed no detectable leaching of metal ions after 2000 cycles.”

Figure R15 | (a-d) Raman spectra of Co₃O₄@SWNT, RuO₂@SWNT, IrO₂@SWNT and SWNT respectively, before and after cycling 2000 times in 0.1 M KOH. Spectra highlight the D band, with intensities normalised to the G band.

11. Bader charge and density of states should be provided by density functional theory calculations.

We thank their reviewer for their suggestion on providing calculated DoS, which are presented below as Figure R16. This data is consistent with other calculations of DoS for filled SWNTs. We have added these new data as Figure S31 and refer to them on page 10 of the main manuscript.

“The calculated density of states of these MO_x@SWNT materials are also shown in Fig. S31.”

Figure R16 | Calculated density of states of (a) Co_3O_4 @SWNT, (b) IrO_2 @SWNT and (c) RuO_2 @SWNT.

In order to quantify the electronic density transfer in the encapsulation complexes, we performed both Mulliken and Bader charge analyses to obtain an integer number of transferred electrons. First, we tested the equivalence of both methods, and we obtained similar results. Hence, all the results shown were obtained using Mulliken charge analysis due to the direct implementation of this method on the CP2K/QUICKSTEP package. We hope this answer provides justification for excluding Bader charge calculations and have added this explanation into the computational details section found on page 3 of the supplementary information for clarity.

12. The language and layout of figures of this manuscript need to be improved.

We thank the review for highlighting the need to improve the document production standard. We have made numerous changes to figures (example, Figure 1, annotations on Figure 2-4) to improve the clarity of presented data. Text throughout, in particular in the introduction, has also been edited to improve clarity.

13. The authors should carefully check the main text to ensure no errors. For example, “the first step in the OER and the RDS (as determined by electrochemical analysis) and may be key to the activity of carbon coated electrocatalysts”, “-OH”, and Refs. 30, 44, 53.

We have endeavoured to correct any mistakes in the main text, including these specific errors that the reviewer has highlighted.

Response to Reviewer 3

This manuscript reports a development of model electrocatalyst with carbon encapsulation to identify catalytic active site in alkaline oxygen evolution reaction (OER). The model catalyst was designed by encapsulating metal oxides (i.e., Ir, Ru, and Co oxides) within single-walled carbon nanotube (SWNT) and capping with fullerene at the end. The synthesized model catalysts showed little response of redox behavior of oxide catalyst, implying successful encapsulation of metal oxide surface by the carbon layer. The prepared catalysts revealed comparable OER activities to the literature without carbon layers and superior stability, based on which the authors argue that carbon surface on metal oxide as major OER active site. Additional in situ Raman spectroscopic analysis and DFT calculation indicate that charge transfer at carbon-metal oxide junction leads to more positively polarized carbon surface during anodic OER condition which can provide beneficial active site for hydroxide adsorption as rate-determining step for OER. The reviewer evaluates this model study approach is well-designed for identification of OER active site. However, several points need to be addressed more thoroughly with major revisions to enhance clarity and improve the quality of manuscript to be suitable for publication in Nature Communications.

We are grateful to the reviewer for their time and the insightful comments on our work. We hope we have addressed these to a high standard.

1. In this manuscript, the authors argue electronic structure modification of carbon surface by encapsulated metal oxide as alkaline OER active site. Although both in situ Raman and DFT results clearly show positively charged carbon surface, the reviewer wonders if the modified electronic structure of carbon layer is comparable to the electronic structure of bare metal oxide surfaces. If not, charge transfer will be dependent on work function difference between carbon layer and metal oxide surface, which possibly implies different effect of carbon layer depending on metal oxide identities. So more general discussion on carbon-metal oxide charge transfer is required in terms of charge transfer.

We agree that the interaction at the interface is critical to address here. Specifically, the interaction between the π -electrons of the carbon nanotube (CNT) and the metal oxide involves both charge transfer and a strong coupling with the delocalized electron cloud on the CNT surface. This interplay results in a significant modification of the electronic structure of the carbon layer, which is not identical to that of bare metal oxides.

The global picture of this interaction is well illustrated in the charge density difference plots and the density of states (DoS) analysis. These results highlight a clear redistribution of charge at the interface and provide evidence for the charge transfer mechanism. The charge density difference plots (for example, Fig R17 (and S28)) demonstrate how the π -electrons of the CNT overlap and interact with the electronic states of the metal oxide, while the DoS calculations (Fig R18) reveal the orbital mixing that

contributes to the modified electronic structure of the system. We have now added the DoS calculation data to the Supporting Information as Figure S31 and highlight this on page 10.

“The calculated density of states of these $MO_x@SWNT$ materials are also shown in Fig. S31.”

Figure R17 | Electronic density difference after the encapsulation of $Co_3O_4@SWNT$ (11,11). Blue (yellow) colour represents the depletion (augmentation) of electronic density.

Figure R18 | Calculated density of states of (a) $Co_3O_4@SWNT$, (b) $IrO_2@SWNT$ and (c) $RuO_2@SWNT$.

2. The SWNT-encapsulated metal oxide catalysts are developed with unique approach. However, its activity comparison to the non-encapsulated (or bare) catalysts are limited in this manuscript and only done by comparison with literature. If the authors first plug fullerene to SWNT and then deposit metal

oxide precursor for catalyst preparation, is it possible to prepare metal oxide catalysts selectively outside of the SWNT? If possible, these catalyst group will provide more clear comparison between OER process on bare MO_x surface and carbon-coated MO_x surface.

We thank the reviewer for the interesting suggestion. In order to get a material with MO_x only on the surface of the nanotubes as a comparison, we conducted tip-closing procedures on our cleaned SWNTs, which involved heating under vacuum at 1000 °C, as demonstrated in the literature.^{16,17} The LSVs of the OER for the resulting materials (*MO_x/C-SWNT*) are presented in Figure R19. The data shows that consistent with our proposed mechanism, the encapsulated metal oxide materials offer similar performance to the catalysts with the metal oxide on the surface of the SWNT. This data is presented within the paper as Figure S15 and is discussed on page 7 of the main manuscript.

“For comparison with our MO_x@SWNT materials, Co₃O₄, RuO₂ and IrO₂ were also deposited onto activated carbons to yield MO_x/AC (see Supplementary Information for details), and OER activities and stabilities were measured (Fig. 2c-e, S15). The initial activities of the MO_x/AC controls were comparable to those of the MO_x@SWNT analogues, but their activities decreased significantly over 2000 cycles. In addition, commercial MO_x catalysts and MO_x deposited onto closed SWNTs (MO_x/C-SWNT) were studied as controls (Fig. S15-S17) and OER activities were similar or lower than those of the MO_x@SWNT materials. Therefore, encapsulation of metal oxides within SWNTs appears to protect the confined species in the MO_x@SWNT materials.”

Figure S19 | a-c) LSVs of Co_3O_4 @SWNT, IrO_2 @SWNT and RuO_2 @SWNT respectively, compared to relevant $\text{MO}_x/\text{C-SWNT}$, MO_x/AC and commercially available MO_x in O_2 saturated 0.1 M potassium hydroxide.

3. In Figure 3f inset and S15, the authors demonstrated suppressed anodic metal oxidation in Co_3O_4 and RuO_2 when encapsulated by SWNT. This result is reasonable in case of complete encapsulation by carbon layer due to the limited access of electrolyte to metal oxide through the carbon layer. However, in this case, it is anticipated that valence state of metal oxide will be different between bare metal oxide and the encapsulated metal oxide at the same anodic potential (i.e., OER operating condition). Have the authors confirmed this with additional analysis such as X-ray absorption spectroscopy? Also, the reviewer wonders how modified metal valence state by the carbon layer can affect OER electrocatalysis compared to the bare metal oxide.

We thank the reviewer for this interesting suggestion. As requested, we have analysed the electronic structure of the encapsulated metal oxide after OER operating conditions, using ex situ XPS after polarisation of the catalyst to 1.6 V vs RHE (Figure R20). Data is shown for plugged RuO_2 @SWNT and IrO_2 @SWNT. Measurements were performed on Co_3O_4 @SWNT but the resulting data displayed poor signal to noise and fitting could not be performed with any confidence (Figure R20a). These data, combined with the electrochemical analysis that shows loss of the standard oxidation peaks, suggests that oxidation of the internal metal oxides still occurs, but at more positive potentials due to the lack of

charge balancing by specifically adsorbing anions. We now present this data as Figure S21 and discuss it on page 8 of the main manuscript.

“It is important to note that some oxidation of the metal oxide was observed using *ex-situ* XPS after polarisation to 1.6 V (Fig. S21), indicating oxidation of the blocked material could occur at higher overpotentials.”

Figure R20 | XPS spectra of a) Co 2p region of Co₃O₄@SWNT b) 4f region of IrO₂@SWNT and c) 3p region of RuO₂@SWNT plugged with C₆₀ after polarisation of 0 to 1.6 V vs RHE in N₂ saturated 0.1 M KOH at 5 mV s⁻¹.

While the change in the XPS spectra of the metal ions suggest a degree of metal oxidation at high overpotentials, the carbon spectra demonstrate the charge transfer from carbon to the metal oxides (Figure R21). This has been added to Figure 1e in the main text, along with discussion on page 4 of the main manuscript.

“XPS of the MO_x@SWNT materials was performed to elucidate the properties of the carbon and the chemical states of the confined MO_x (Fig. 1e and S10 and 11). To accurately compare the position of the carbon 1s peak, the samples were loaded onto silicon wafers, and the Si 2p peak at 99.5 eV⁶⁴ was used as an internal reference to correct for charge accumulation (Fig. S10). The carbon peaks of all MO_x@SWNT were shifted to a higher binding energy compared to those of unfilled SWNTs, with RuO₂ displaying the largest shift. This shift suggests that electron density on the carbon surface decreased due to charge transfer to the encapsulated metal oxide, which is consistent with reports of similar materials.⁶³”

Figure R21 | X-ray photoelectron spectra showing the C 1s region of RuO₂@SWNT, IrO₂@SWNT, Co₃O₄@SWNT and SWNT. Inset shows the highlighted peak maxima. Wide-scan spectra of each material can be found in the SI (Fig. S10).

This supports our conclusions presented above where charge density difference plots demonstrate how the π -electrons of the CNT overlap and interact with the electronic states of the metal oxide, while the DoS calculations reveal the orbital mixing that contributes to the modified electronic structure of the system. These phenomena make the hydroxide binding events more favourable.

4. In ex situ Raman analysis (Figure S12), a blueshift of the G-band peak does not happen in Co₃O₄@SWNT in contrary to RuO₂@SWNT and IrO₂@SWNT. The author indicates that the charge transfer is less significant for Co₃O₄@SWNT. However, in situ Raman shift of G band position and DFT results are provided as a proof of modified electronic structure on the carbon surface for Co₃O₄@SWNT at the same time. This discrepancy should be further discussed in detail to fully support the authors' argument.

We thank the reviewer for bringing this to our attention. This discrepancy is due to poor spectral resolution in our initial setup, with the G band shift of Co₃O₄@SWNT compared to SWNT within the resolution. In an effort to clarify this we have performed Raman spectroscopy again, this time using a higher grating of 1800 mm⁻¹, giving spectral resolution of 0.43 cm⁻¹ as compared to 1.8 cm⁻¹. We have also averaged 25 spectra across a 50 μ m x 50 μ m region to make things more representative for each sample. Using these new parameters, a clear trend in the position of the G band can be seen in each material (Figure. R22), which follows the stepwise magnitude of charge transfer shifts we have calculated computationally. This data has been added to the main paper within Figure 1f and is discussed on page 5.

“The G-band positions in the spectra of all MO_x were blueshifted relative to that of unfilled SWNT (Fig. 1f), with RuO₂@SWNT displaying the largest shift, followed by IrO₂@SWNT and then Co₃O₄@SWNT. These

shifts were in line with XPS analysis and further evidence of charge transfer from the SWNT to the encapsulated metal oxides.⁶⁶ ”

Figure R22 | Raman spectra showing the G band of RuO₂@SWNT, IrO₂@SWNT, Co₃O₄@SNWT and SWNT (a) and highlighted G band maxima (b).

5. In Figure 3, OER activities are compared between before and after fullerene plugging in MOx@CWNT. While RuO₂ and IrO₂ catalysts clearly show negligible change through the fullerene plugging, Co₃O₄@SWNT catalyst shows a distinct increase in OER overpotential in overall current range. Does this activity loss come from loss of Co₃O₄ surface at the end of Co₃O₄@CWNT structure? If it is, does it indicate much higher TOF of Co₃O₄ surface exposed at the end of Co₃O₄@SWNT structure compared to the Co₃O₄ surface encapsulated by CWNT? Moreover, why this feature occurs only in Co₃O₄@CWNT compared to RuO₂@SWNT and IrO₂@SWNT? Plus, why does the plugged Co₃O₄@SWNT show decreased OER activity at low current density (below ~2 mA cm⁻²) compared to the bare SWNT?

To understand the impact of the fullerenes and the plugging on the OER, we have measured the OER activity of bare SWNTs and of fullerene-plugged SWNTs by LSV (Figure R23). We found that the fullerenes had little effect on the OER activity. This data has been added as Figure S23 and we have updated the discussion on the implications of plugging on page 8 of the main manuscript.

“Finally, to test whether fullerene encapsulation has an inherent effect on the OER activity of the carbon shell, we plugged pristine SWNTs with fullerenes. No significant change in OER activity was observed after plugging (Fig. S23).”

Figure R23 | RDE-LSVs at SWNT and C₆₀-SWNT in N₂ saturated 0.1 M potassium hydroxide at 5 mV s⁻¹ and 1600 rpm.

Regarding the important points highlighted by the reviewer, it's possible that the drop in activity seen in the Co₃O₄ and IrO₂ systems may be due to the plugging blocking access of the exposed metal oxide at the ends of the SWNT (decreasing the accessible surface area of the catalyst). We have also seen that the process of plugging the tubes with fullerenes can lead to an overall reduction in electrochemically active surface area.¹⁸

6. In addition to the 5th comment, it is necessary to show OER activity of fullerene-plugged SWNT (C60@SWNT) without metal oxide as comparison to confirm negligible role of fullerene in OER activity (likely in Supporting Information).

Please see our answer to the comment above, which we hope answers this in detail.

7. Have the authors applied this approach to acidic OER electrocatalysis (such as IrO₂@SWNT)? The reviewer wonders if there is any change in electrocatalytic mechanism between adsorbate evolution mechanism and lattice oxygen evolution mechanism.

In this study we have chosen to focus on OER in alkaline conditions due to a simplicity in the system to prove that the carbon surface is the location of the active site. Specifically, it has been shown previously that protons can travel through SWNT sidewalls,¹⁸ which would drastically alter the study. However, we do believe that this is an important area to explore, and endeavour to investigate this in further studies. We discuss our motivation for performing these studies in alkaline conditions on page 6 of the main manuscript.

"We selected alkaline rather than acidic conditions as a starting point for this study based on our recent observation that protons can transport readily through the graphenic walls of SWNTs,⁷⁰ potentially complicating the electrochemical reaction mechanism."

- 1 L. Guan, Z. Shi, M. Li and Z. Gu, *Carbon*, 2005, **43**, 2780–2785.
- 2 W. J. Cull, S. T. Skowron, R. Hayter, C. T. Stoppiello, G. A. Rance, J. Biskupek, Z. R. Kudrynskyi, Z. D. Kovalyuk, C. S. Allen, T. J. A. Slater, U. Kaiser, A. Patanè and A. N. Khlobystov, *ACS Nano*, 2023, **17**, 6062–6072.
- 3 C. T. Stoppiello, J. Biskupek, Z. Y. Li, G. A. Rance, A. Botos, R. M. Fogarty, R. A. Bourne, J. Yuan, K. R. J. Lovelock, P. Thompson, M. W. Fay, U. Kaiser, T. W. Chamberlain and A. N. Khlobystov, *Nanoscale*, 2017, **9**, 14385–14394.
- 4 S. Dou, C.-L. Dong, Z. Hu, Y.-C. Huang, J. Chen, L. Tao, D. Yan, D. Chen, S. Shen, S. Chou and S. Wang, *Adv. Funct. Mater.*, 2017, **27**, 1702546.
- 5 B. Jiang, X. Fan, Q. Dang, F. Liao, Y. Li, H. Lin, Z. Kang and M. Shao, *Nano Energy*, 2020, **76**, 105079.
- 6 I. Udrea and S. Avramescu, *Environ. Technol.*, 2004, **25**, 1131–1141.
- 7 J.-H. Jang, A. A. Jeffery, J. Min, N. Jung and S. J. Yoo, *Nanoscale*, 2021, **13**, 15116–15141.
- 8 T. Naito, T. Shinagawa, T. Nishimoto and K. Takanebe, *Inorg. Chem. Front.*, 2021, **8**, 2900–2917.
- 9 S. Jung, C. C. L. McCrory, I. M. Ferrer, J. C. Peters and T. F. Jaramillo, *J. Mater. Chem. A*, 2016, **4**, 3068–3076.
- 10 T. Shinagawa, A. T. Garcia-Esparza and K. Takanebe, *Sci. Rep. 2015 51*, 2015, **5**, 1–21.
- 11 Y. Cheng, F. Kwofie, Z. Chen, R. Zhang, Z. Wang, S. P. Jiang, J. Zheng and H. Tang, *Electrochimica Acta*, 2023, **438**, 141146.
- 12 D. S. Jensen, S. S. Kanyal, N. Madaan, M. A. Vail, A. E. Dadson, M. H. Engelhard and M. R. Linford, *Surf. Sci. Spectra*, 2013, **20**, 36–42.
- 13 K. R. Kissell, K. B. Hartman, P. A. W. Van der Heide and L. J. Wilson, *J. Phys. Chem. B*, 2006, **110**, 17425–17429.
- 14 P. Grinberg, R. E. Sturgeon, L. de O. Diehl, C. A. Bizzi and E. M. M. Flores, *Spectrochim. Acta Part B At. Spectrosc.*, 2015, **105**, 89–94.
- 15 J. W. Jordan, K. L. Y. Fung, S. T. Skowron, C. S. Allen, J. Biskupek, G. N. Newton, U. Kaiser and A. N. Khlobystov, *Chem. Sci.*, 2021, **12**, 7377–7387.
- 16 H. Z. Geng, X. B. Zhang, S. H. Mao, A. Kleinhammes, H. Shimoda, Y. Wu and O. Zhou, *Chem. Phys. Lett.*, 2004, **399**, 109–113.
- 17 S. Bandow, M. Takizawa, K. Hirahara, M. Yudasaka and S. Iijima, *Chem. Phys. Lett.*, 2001, **337**, 48–54.
- 18 J. W. Jordan, B. Mortiboy, A. N. Khlobystov, L. R. Johnson, G. N. Newton and D. A. Walsh, *J. Am. Chem. Soc.*, 2023, **145**, 9052–9058.